# A terminal selector prevents a Hox transcriptional switch to safeguard motor neuron identity throughout life

Weidong Feng[1,2], Yinan Li[1,3], Pauline Dao[1], Jihad Aburas[1], Priota Islam[4,5], Benayahu Elbaz[6], Anna Kolarzyk[6], André EX Brown[4,5], Paschalis Kratsios[1,2,3,7]*

[1]Department of Neurobiology, University of Chicago, Chicago, United States; [2]Committee on Development, Regeneration and Stem Cell Biology, University of Chicago, Chicago, United States; [3]Committee on Neurobiology, University of Chicago, Chicago, United States; [4]MRC London Institute of Medical Sciences, London, United Kingdom; [5]Institute of Clinical Sciences, Imperial College London, London, United Kingdom; [6]Department of Neurology, Center for Peripheral Neuropathy, University of Chicago, Chicago, United States; [7]The Grossman Institute for Neuroscience, Quantitative Biology, and Human Behavior, University of Chicago, Chicago, United States

**Abstract** To become and remain functional, individual neuron types must select during development and maintain throughout life their distinct terminal identity features, such as expression of specific neurotransmitter receptors, ion channels and neuropeptides. Here, we report a molecular mechanism that enables cholinergic motor neurons (MNs) in the *C. elegans* ventral nerve cord to select and maintain their unique terminal identity. This mechanism relies on the dual function of the conserved terminal selector UNC-3 (Collier/Ebf). UNC-3 synergizes with LIN-39 (Scr/Dfd/Hox4-5) to directly co-activate multiple terminal identity traits specific to cholinergic MNs, but also antagonizes LIN-39's ability to activate terminal features of alternative neuronal identities. Loss of *unc-3* causes a switch in the transcriptional targets of LIN-39, thereby alternative, not cholinergic MN-specific, terminal features become activated and locomotion defects occur. The strategy of a terminal selector preventing a transcriptional switch may constitute a general principle for safeguarding neuronal identity throughout life.

*For correspondence:
pkratsios@uchicago.edu

**Competing interests:** The authors declare that no competing interests exist.

## Introduction

Every nervous system is equipped with distinct neuron types essential for different behaviors. Fundamental to nervous system function is the precise establishment and maintenance of neuron type-specific gene expression programs. Integral components of such programs are effector genes that encode proteins critical for neuronal function (e.g., neurotransmitter [NT] biosynthesis components, ion channels, NT receptors, neuropeptides) (*Deneris and Hobert, 2014*; *Hobert, 2008*; *Hobert, 2011*; *Hobert, 2016*). These effector genes, referred to as terminal identity genes herein, are expressed continuously, from development throughout life, in post-mitotic neurons in a combinatorial fashion (*Hobert, 2008*). Hence, it is the unique overlap of many effector gene products in a specific neuron type that determines its distinct terminal identity, and thereby function. However, the molecular mechanisms that select, in individual neuron types, which terminal identity genes should be expressed and which ones should be repressed are poorly defined. Understanding how neuron type-specific batteries of terminal identity genes are established during development and, perhaps most importantly, maintained throughout life represents one key step towards

understanding how individual neuron types become and remain functional. Providing molecular insights into this fundamental problem may also have important biomedical implications, as defects in terminal identity gene expression are associated with a variety of neurodevelopmental and neuro-degenerative disorders (*Deneris and Hobert, 2014*; *Shibuya et al., 2011*; *Imbrici et al., 2013*; *Sgadò et al., 2011*).

Seminal genetic studies in multiple model systems revealed a widely employed molecular principle: neuron type-specific transcription factors (TFs) often coordinate the expression of 'desired' terminal identity genes with the exclusion of 'unwanted' terminal identity genes (*Morey et al., 2008*; *Sagasti et al., 1999*; *Britanova et al., 2008*; *Cheng et al., 2004*; *Kala et al., 2009*; *Lopes et al., 2012*; *Mears et al., 2001*; *Nakatani et al., 2007*). These TFs exert a dual role: they are not only required to induce a specific set of terminal identity features critical for the function of a given neuron type, but also to simultaneously prevent expression of molecular features normally reserved for other neuron types. Consequently, neurons lacking these TFs fail to acquire their unique terminal identity, and concomitantly gain features indicative of alternative identities. For example, mouse striatal cholinergic interneurons lacking *Lhx7* lose their terminal identity and acquire molecular features indicative of GABAergic interneuron identity (*Lopes et al., 2012*). In midbrain neurons, removal of *Gata2* results in loss of GABAergic identity and simultaneous gain of terminal identity features specific to glutamatergic neurons (*Kala et al., 2009*). However, the molecular mechanisms underlying the dual function of most neuron type-specific TFs remain poorly defined. How can the same TF, within the same cell, promote a specific identity and simultaneously prevent molecular features of alternative neuronal identities? In principle, the same TF can simultaneously operate as direct activator of neuron type-specific terminal identity genes and direct repressor of alternative identity genes (*Lodato et al., 2014*; *Wyler et al., 2016*). Another possibility is indirect regulation. For example, a neuron type-specific TF can prevent adoption of alternative identity features by repressing expression of an intermediary TF that normally promotes such features (*Cheng et al., 2004*). Other mechanisms involving TF competition for cell type-specific enhancers or cell type-specific TF-TF interactions have also been described (see Discussion) (*Andzelm et al., 2015*; *Gordon and Hobert, 2015*; *Rhee et al., 2016*; *Thaler et al., 2002*). It remains unclear, however, whether these mechanisms of action of neuron type-specific TFs are broadly applicable in the nervous system.

Although the aforementioned studies begin to explain how neurons select their terminal identity features during development (*Morey et al., 2008*; *Sagasti et al., 1999*; *Britanova et al., 2008*; *Cheng et al., 2004*; *Kala et al., 2009*; *Lopes et al., 2012*; *Mears et al., 2001*; *Nakatani et al., 2007*), the function of neuron type-specific TFs is rarely assessed during post-embryonic stages. Hence, the molecular mechanisms that maintain neuronal terminal identity features, and thereby neuronal function, are largely unknown. Is the same neuron type-specific TF continuously required, from development through adulthood, to induce a specific set of terminal identity genes and simultaneously prevent 'unwanted' features? Alternatively, a given neuron type could employ different mechanisms for selection (during development) and maintenance (through adulthood) of its function-defining terminal features. Addressing this fundamental problem has been challenging in the vertebrate nervous system, in part due to its inherent complexity and difficulty to track individual neuron types with single-cell resolution from embryo to adult.

To study how neurons select and maintain their terminal identity features, we use as a model the well-defined motor neuron (MN) subtypes of the *Caenorhabditis elegans* ventral nerve cord (equivalent to vertebrate spinal cord). Five cholinergic (DA, DB, VA, VB, AS) and two GABAergic (DD, VD) MN subtypes are located along the nerve cord and control locomotion (*Figure 1A*) (*Von Stetina et al., 2006*; *White et al., 1986*). Because they are present in both *C. elegans* sexes (males and hermaphrodites), we will refer to them as 'sex-shared' MNs. In addition, there are two subtypes of 'sex-specific' cholinergic MNs: the hermaphrodite-specific VC neurons control egg laying (*Portman, 2017*; *Schafer, 2005*), and the male-specific CA neurons are required for mating (*Schindelman et al., 2006*) (*Figure 1A*). In addition to distinct morphology and connectivity, each subtype can be molecularly defined by the combinatorial expression of known terminal identity genes, such as ion channels, NT receptors, and neuropeptides (*Figure 1B*). An extensive collection of transgenic reporter *C. elegans* animals for MN subtype-specific terminal identity genes is available, thereby providing a unique opportunity to investigate, at single-cell resolution, the effects of TF gene removal on developing and adult MNs.

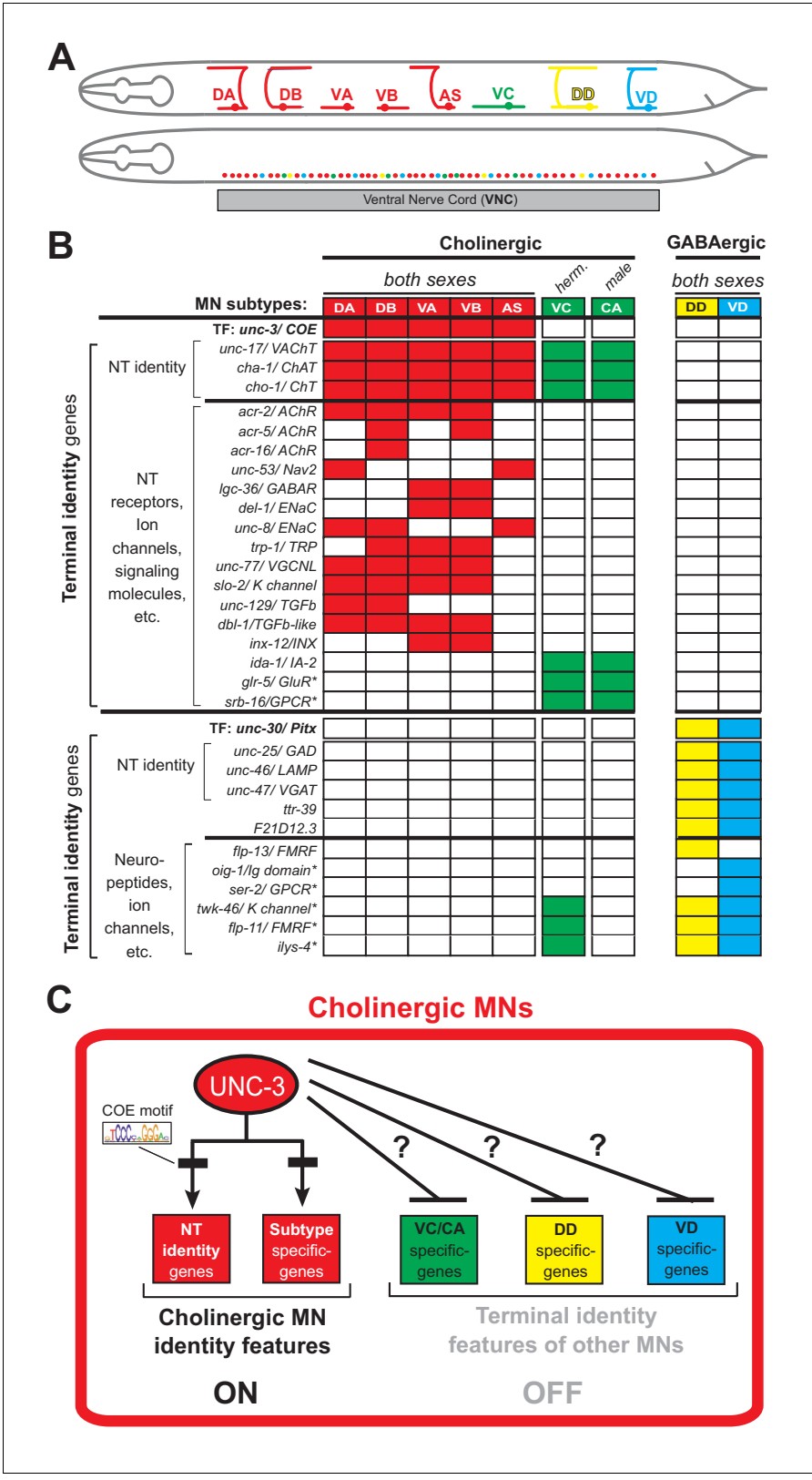

**Figure 1.** An extensive collection of terminal identity markers for distinct motor neuron subtypes of the *C. elegans* ventral nerve cord. (**A**) Schematic showing distinct morphology for each motor neuron subtype in the *C. elegans* hermaphrodite. Below, colored dots represent the invariant cell body position of all MNs of the ventral nerve cord (VNC). Red: 39 sex-shared cholinergic MNs (DA2–7 = 6 neurons, DB3–7 = 5, VA2–11 = 10, VB3–11 = 9,

*Figure 1 continued*

AS2−10 = 9); Green: six hermaphrodite-specific VC MNs; Yellow: four sex-shared GABAergic DD neurons (DD2−5 = 4); Blue: nine sex-shared GABAergic VD neurons (VD3−11 = 9). With the exception of VC, all other subtypes have 1–3 extra neurons located at the flanking ganglia (retrovesicular and pre-anal) of the VNC (not shown). Individual neurons of each subtype intermingle along the VNC. (B) Table summarizing expression of terminal identity markers for VNC MNs. The sex-shared GABAergic MNs (DD, VD) and the sex-specific MNs (VC, CA) do not express UNC-3. Conversely, the sex-shared cholinergic MNs (DA, DB, VA, VB, AS) and the sex-specific MNs (VC, CA) do not express UNC-30/Pitx. For the genes indicated with an asterisk (*), a detailed expression pattern is provided in *Figure 1—figure supplement 1*. Of note, the male-specific MNs of the CP subtype are also not shown. (C) Schematic that summarizes the known function of UNC-3 (activator of cholinergic MN identity genes) and the question under investigation: does UNC-3 prevent expression of terminal identity features reserved for other MN subtypes?.

The online version of this article includes the following figure supplement(s) for figure 1:

**Figure supplement 1.** Detailed characterization of the expression pattern of VC and VD terminal identity markers.

---

UNC-3, the sole *C. elegans* ortholog of the Collier/Olf/Ebf (COE) family of TFs, is selectively expressed in all sex-shared cholinergic MNs of the nerve cord (*Figure 1B*) (*Kratsios et al., 2017*; *Kratsios et al., 2012*; *Pereira et al., 2015*; *Prasad et al., 2008*; *Prasad et al., 1998*). Animals lacking *unc-3* display striking locomotion defects (*Brenner, 1974*). UNC-3 is known to directly activate a large battery of terminal identity genes expressed either in all sex-shared cholinergic MNs (e.g., the NT identity genes *unc-17/* VAChT and *cha-1/* ChAT), or in certain subtypes (e.g., ion channels, NT receptors, signaling molecules) (*Kratsios et al., 2012*) (*Figure 1B–C*). Based on its ability to broadly co-regulate many distinct terminal identity features, *unc-3* has been classified as a terminal selector gene (*Hobert, 2008*). Besides its well-established function as activator of terminal identity genes in cholinergic MNs, whether and how UNC-3 can prevent expression of terminal features of alternative neuronal identities remains unclear.

Here, we describe a dual role for UNC-3 that enables sex-shared cholinergic MNs to select during development and maintain throughout life their terminal identity features. We find that UNC-3 is continuously required - from development through adulthood - not only to activate cholinergic MN identity genes, but also to prevent expression of terminal features normally reserved for other MN subtypes of the nerve cord, namely the sex-shared GABAergic VD neurons and sex-specific cholinergic MNs (CA, VC). These findings lend support to the notion that neuron type-specific TFs can promote a specific identity and simultaneously suppress features reserved for alternative, but functionally related, neuronal identities.

To uncover the molecular mechanism underlying the dual role of UNC-3, we conducted an unbiased genetic screen, which led to the identification of the Hox protein LIN-39 (Scr/Dfd/Hox4-5) as the intermediary factor necessary for expression of alternative neuronal identity features (e.g., VD, VC) in *unc-3*-depleted MNs. Unlike previously described cases of TFs that act indirectly to prevent alternative neuronal identities by repressing intermediary factors (discussed earlier), UNC-3 does not repress *lin-39* and both factors are co-expressed in cholinergic MNs. However, UNC-3 antagonizes the ability of LIN-39 to induce terminal features of alternative identities. Intriguingly, UNC-3 also synergizes with LIN-39 to co-activate multiple terminal identity features specific to cholinergic MNs. Consequently, loss of *unc-3* causes a switch in the transcriptional targets of LIN-39, thereby alternative, not cholinergic MN-specific, terminal identity features become activated and locomotion defects occur. Given that terminal selectors and Hox proteins are expressed in a multitude of neuron types across species (*Deneris and Hobert, 2014*; *Hobert and Kratsios, 2019*; *Philippidou and Dasen, 2013*; *Estacio-Gómez and Díaz-Benjumea, 2014*), the strategy of a terminal selector preventing a Hox transcriptional switch may constitute a general principle for safeguarding neuronal identity throughout life.

## Results

### UNC-3 has a dual role in distinct populations of ventral nerve cord (VNC) motor neurons

Neuron type-specific TFs often promote a specific identity and simultaneously suppress features reserved for other, functionally related neuronal types (*Arlotta and Hobert, 2015*). To test this notion for UNC-3, it was essential to identify a set of terminal identity markers for all *unc-3*-negative MN subtypes of the VNC, namely the GABAergic (VD, DD) and sex-specific (VC, CA) MNs (*Figure 1B*). We undertook a candidate gene approach and examined the precise expression pattern of terminal identity genes (e.g., NT receptors, signaling proteins, ion channels, neuropeptides) reported to be expressed in *unc-3*-negative MNs (www.wormbase.org). In total, we carefully characterized at single-cell resolution the expression of 15 genes in wild-type animals of both *C. elegans* sexes at the fourth larval stage (L4) (see Materials and methods and *Figure 1—figure supplement 1*). This analysis provided nine terminal identity markers highly specific to *unc-3*-negative MNs that fall into four categories (*Figure 1B*): (a) two VD-specific markers (*ser-2*/serotonin receptor [ortholog of HTR1D]; *oig-1*/ one Ig domain protein), (b) one DD-specific marker (*flp-13*/ FMRF like neuropeptide), (c) three markers for sex-specific (VC in hermaphrodites, CA in males) MNs (*glr-5*/glutamate receptor [ortholog of GRID/GRIK]; *srb-16*/serpentine GPCR receptor; *ida-1*/ortholog of protein tyrosine phosphatase PTPRN), and (**d**) three markers expressed in both GABAergic subtypes (DD, VD) and sex-specific MNs (*flp-11*/FRMR like neuropeptide, *twk-46*/potassium channel [ortholog of KCNK1], *ilys-4/* invertebrate type lysozyme).

These nine markers enabled us to test whether *unc-3*-depleted MNs gain expression of terminal features normally reserved for other MN subtypes. By using animals carrying a strong loss-of-function (null) allele for *unc-3 (n3435)* (*Prasad et al., 2008*), we first assessed any putative effects on terminal markers for the sex-shared GABAergic MNs (DD, VD). Although the DD-specific marker *flp-13* is unaffected (*Figure 2—figure supplement 1*, panel **A**), ectopic expression of the VD-specific markers (*ser-2*, *oig-1*) was observed in *unc-3*-depleted MNs (*Figure 2A–B*). Interestingly, this ectopic expression was region-specific, observed in cholinergic MNs of the mid-body region of the VNC with 100% penetrance (*Figure 2A–B*). Importantly, 12.1 ± 2.6 (mean ± STDV) out of the 39 *unc-3*-depleted MNs in the VNC were ectopically expressing these VD markers, suggesting that not all *unc-3*-depleted MNs acquire VD terminal identity features. Given that GABAergic and cholinergic MNs are generated in normal numbers in *unc-3* animals (*Kratsios et al., 2012*), the increase in the number of neurons expressing the VD markers cannot be attributed to early developmental defects affecting MN numbers. We next asked whether these ~12 MNs adopt additional VD terminal identity features, such as expression of genes involved in GABA biosynthesis (*unc-25/GAD* and *unc-47/ VGAT)*, or selectively expressed in GABAergic MNs (*ttr-39, klp-4*). However, this does not appear to be the case, arguing against a complete cell fate switch (*Figure 2—figure supplement 1*, panel A). We conclude that, in the absence of *unc-3,* cholinergic MNs not only lose their original terminal identity, but a third of them (~12 out of 39) in the mid-body VNC region also gain some terminal identity features normally reserved for the sex-shared VD neurons (*Figure 1*). We will refer to these *unc-3*-depleted MNs as 'VD-like' (*Figure 2G*). We also uncovered the identity of these cells across multiple *unc-3* mutant animals and conclude that it is the same 12 neurons that become VD-like across animals (*Figure 2—figure supplement 1*, panel B).

To test whether UNC-3 also prevents expression of terminal identity features of sex-specific cholinergic MNs, we examined three VC-specific terminal markers (*glr-5, srb-16, ida-1* in *Figure 1B*) in hermaphrodite nematodes lacking *unc-3*. Again, we observed region-specific effects with 100% penetrance in the same cells across multiple animals (*Figure 2C–D*, *Figure 2—figure supplement 1*, panel B). All three markers were ectopically expressed in 10.5 ± 3.7 (mean ± STDV) of the 39 *unc-3*-depleted MNs located in the mid-body region of the VNC (*Figure 2C–D*). These results are in agreement with a previous study reporting ectopic *ida-1* expression in *unc-3*-depleted MNs (*Prasad et al., 2008*). If these ~11 MNs fully adopt the VC terminal identity, then they should also express genes necessary for acetylcholine biosynthesis since VC neurons are cholinergic. However, this is not the case as expression of *unc-17*/VAChT and *cho-1*/ChT is dramatically affected in *unc-3*-depleted MNs (*Kratsios et al., 2012*). These data suggest that ~11 of the 39 *unc-3*-depleted MNs in the mid-body VNC region adopt some, but not all, VC terminal identity features. We will therefore refer to these *unc-3*-depleted MNs as 'VC-like' (*Figure 2G*).

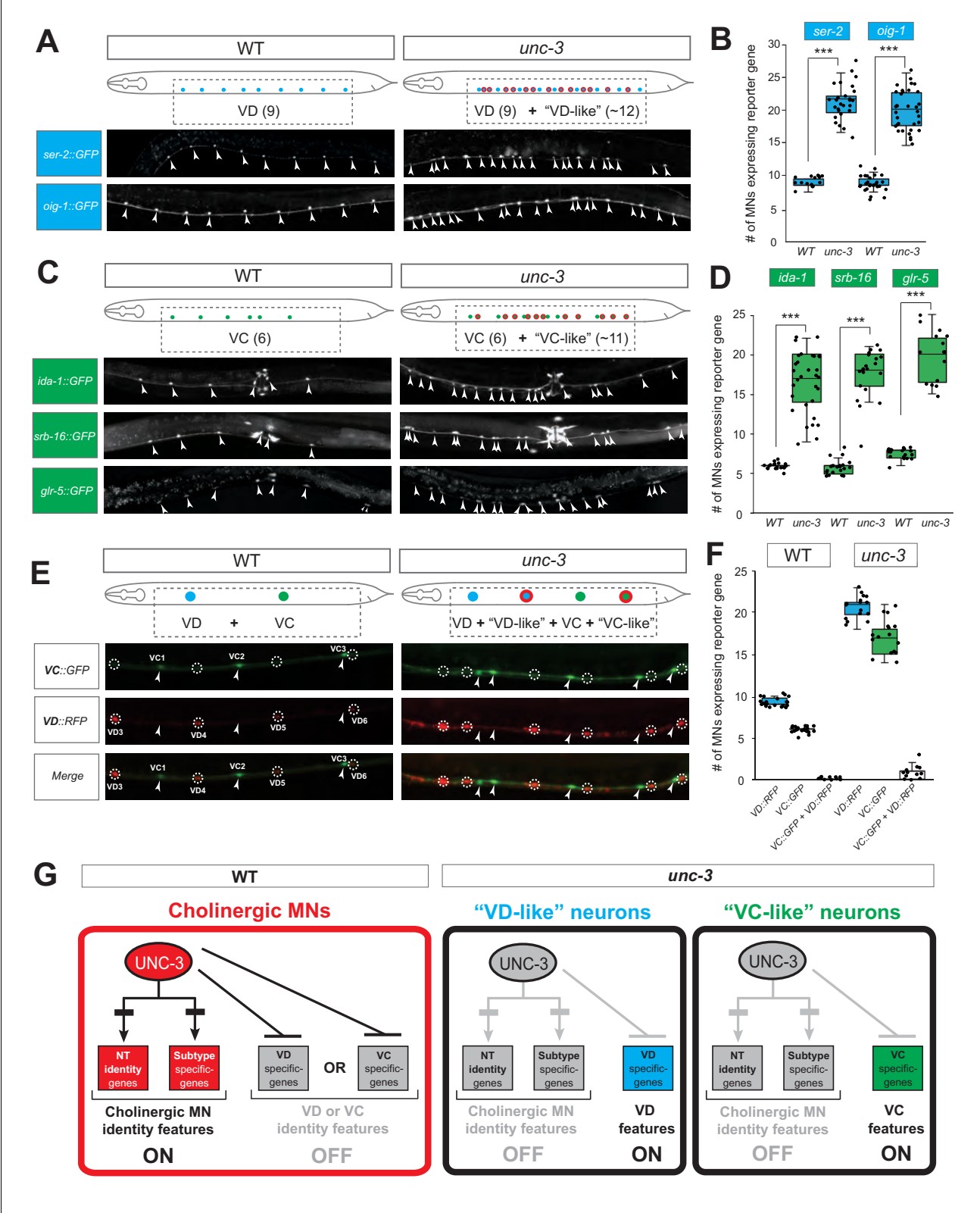

**Figure 2.** UNC-3 has a dual role in cholinergic ventral cord motor neurons. (**A**) Terminal identity markers of VD neurons (*ser-2, oig-1*) are ectopically expressed in *unc-3*-depleted MNs. Representative images of larval stage 4 (L4) hermaphrodites are shown. Similar results were obtained in adult animals. Arrowheads point to MN cell bodies with *gfp* marker expression. Green fluorescence signal is shown in white for better contrast. Dotted black box indicates imaged area. (**B**) Quantification of VD markers (*ser-2, oig-1*) in WT and *unc-3 (n3435)* at L4. N > 15. ***p<0.001. For details on box plots,

*Figure 2 continued on next page*

*Figure 2 continued*

see Materials and methods. (C) Terminal identity markers of VC neurons (*ida-1, srb-16, glr-5*) are ectopically expressed in *unc-3*-depleted MNs. Representative images of larval stage 4 (L4) hermaphrodites are shown. Similar results were obtained in adult animals. Arrowheads point to MN cell bodies with *gfp* marker expression. Green fluorescence signal is shown in white for better contrast. Dotted black box indicates imaged area. (D): Quantification of VC markers (*ida-1, srb-16, glr-5*) in WT and *unc-3 (n3435)* at L4. Individual data points are dot-plotted. N > 15. ***p<0.001. (E) Distinct MNs acquire VC-like or VD-like terminal identity features in *unc-3 (n3435)* mutants. The VC marker in green (*ida-1::gfp*) and the VD marker in red (*ser-2::rfp*) do not co-localize in WT or *unc-3 (n3435)* mutants. Representative images are shown. Individual VC/VC-like and VD/VD-like neurons are pointed and circled, respectively,(VD: dotted circles; VC: arrowheads) to highlight that an individual MN never expresses both markers. (F) Quantification of data shown in E. N > 16. (G) Schematic that summarizes the dual role of *unc-3*. Apart from activating cholinergic MN terminal identity genes, UNC-3 prevents expression of VD and VC terminal features in distinct cells ('VD-like' versus 'VC-like').

The online version of this article includes the following figure supplement(s) for figure 2:

**Figure supplement 1.** UNC-3 selectively prevents expression of VD and VC terminal identity features in distinct cholinergic MNs.

**Figure supplement 2.** The dual role of UNC-3 in cholinergic MNs extends to both *C. elegans* sexes.

Are the VD-like and VC-like neurons in *unc-3* hermaphrodites distinct populations? To test this, we generated *unc-3* hermaphrodites that carry a green fluorescent reporter for VC terminal identity (*ida-1::gfp*) and a red reporter for VD identity (*ser-2::rfp*). We found no overlap of the two reporters, indicating that the VD-like and VC-like neurons represent two distinct populations (*Figure 2E–F*). We further corroborated this result by taking advantage of the invariant lineage and cell body position of all MNs along the *C. elegans* nerve cord (*Figure 2—figure supplement 1*, panel B). Of note, the VC-like population appears to be lineally related to VC neurons, whereas the VD-like population is not lineally related to VD neurons (*Figure 2—figure supplement 1*, panels B-C). Lastly, terminal identity markers normally expressed in both VD and VC neurons (*flp-11, ilys-4, twk-46*) display an additive effect in *unc-3* mutants, as they are ectopically expressed in both VD-like and VC-like populations, further suggesting the presence of distinct *unc-3* MN populations (*Figure 2—figure supplement 1*, panels D-E).

To summarize, there are 39 *unc-3*-expressing MNs along the wild-type nerve cord in hermaphrodites. While loss of *unc-3* uniformly leads to loss of cholinergic identity in all these MNs (*Kratsios et al., 2012*), one population (~12 MNs) acquires VD-like molecular features, while a second population (~11 MNs) acquires VC-like molecular features, uncovering a dual role of UNC-3 in these populations (*Figure 2G*). Of note, the remaining MNs (~16) in the VNC of *unc-3* mutants [39 - (12 VD-like + 11 VC-like)=16] do not gain either VD or VC terminal identity features.

## The dual role of UNC-3 in cholinergic MNs extends to both *C. elegans* sexes

To test whether the dual function of UNC-3 applies to both sexes, we extended our analysis to *C. elegans* males. First, we showed that loss of *unc-3* in males resulted in loss of several cholinergic MN terminal identity features (*Figure 2—figure supplement 2*). Second, we observed ectopic expression of VD-specific terminal identity markers (*oig-1, ser-2*) in 11.9 ± 3.9 (mean ± STDV) out of the 39 *unc-3*-depleted MNs, indicating the presence of 'VD-like' neurons in the male nerve cord (*Figure 2—figure supplement 2*). Lastly, we asked whether *unc-3* loss leads to ectopic expression of terminal identity markers (*ida-1, srb-16, glr-5*) for male-specific CA neurons. Indeed, we found this to be the case (*Figure 2—figure supplement 2*), suggesting the adoption of 'CA-like' features by a population of *unc-3*-depleted MNs. Similar to hermaphrodites, these VD-like and CA-like cells were observed in the mid-body region of the male nerve cord with 100% penetrance (*Figure 2—figure supplement 2*).

Taken together, our findings uncover a dual role for UNC-3 in sex-shared cholinergic MNs. UNC-3 is not only required to activate cholinergic MN identity genes (*Kratsios et al., 2012*), but also to prevent expression of molecular features normally reserved for three other, functionally related neuronal subtypes of the nerve cord (VD, VC, CA). In both sexes, UNC-3 prevents expression of select terminal features of VD neurons in a specific population of cholinergic MNs. In a second population, UNC-3 prevents expression of terminal features normally reserved for sex-specific MNs, that is VC features in hermaphrodites and CA features in males. In the ensuing sections, we focus our analysis on *C. elegans* hermaphrodites to dissect the molecular mechanism underlying the dual role of UNC-3.

## UNC-3 is continuously required to prevent expression of VD and VC terminal identity features

Neuron type-specific TFs that promote a specific identity and simultaneously prevent alternative features have been previously described (see Introduction). However, whether this dual role is required transiently (during development), or continuously (throughout life) remains unclear. The UNC-3 case provides an opportunity to distinguish between these two possibilities because ectopic expression of VC and VD features is observed at both larval and adult stages in *unc-3* null animals (*Figure 2*, *Figure 2—figure supplement 1*). To this end, we employed the auxin-inducible degron (AID) system that enables depletion of UNC-3 in a temporally controlled manner (*Zhang et al., 2015*). This system requires tagging the UNC-3 protein with the AID degron fused to a fluorescent reporter gene (mNeonGreen, mNG). When UNC-3::mNG::AID and the plant-specific F-box protein TIR1 are co-expressed in MNs (by crossing animals carrying the *unc-3::mNG::AID* allele with *eft-3::TIR1* transgenic animals), application of the plant hormone auxin on these double transgenic animals induces degradation of UNC-3::mNG::AID (*Figure 3A–C*). Auxin administration at the L4 stage (last larval stage before adulthood) on *unc-3::mNG::AID; eft-3::TIR1* animals resulted in a dramatic depletion of UNC-3 at day one adult animals (24 hr after auxin). UNC-3 depletion was accompanied by ectopic expression of VD and VC terminal identity features in nerve cord MNs, demonstrating a post-embryonic requirement for UNC-3 (*Figure 3D–E*). Similar results were obtained when auxin was applied at different time points (*Figure 3D*, legend). These findings suggest that UNC-3 is continuously required to prevent expression of VD and VC terminal identity features.

## UNC-3 acts indirectly to prevent expression of VD and VC terminal identity genes

How does UNC-3 activate cholinergic MN identity genes and simultaneously prevent terminal features of alternative MN identities (e.g., VD, VC) (*Figure 2G*)? Based on previous reports, the same TF, within the same neuron, can act as a direct activator for a set of genes and a direct repressor for another set of genes (*Lodato et al., 2014*; *Wyler et al., 2016*; *Borromeo et al., 2014*). While it is known that UNC-3 acts directly – through its cognate binding site (COE motif) – to activate expression of a large battery of cholinergic MN identity genes, we did not find any COE motifs in the *cis*-regulatory region of VD or VC terminal identity genes (*Supplementary file 1*). This contrasts the previously described function of UNC-3 as direct repressor (through the COE motif) of terminal identity genes in the chemosensory ASI neurons of *C. elegans* (*Kim et al., 2005*).

To test the possibility of indirect repression via an intermediary factor, we focused on VD neurons because, unlike VC neurons, a known activator of VD terminal features has been reported (*Cinar et al., 2005*; *Eastman et al., 1999*; *Jin et al., 1994*). In wild-type animals, the TF UNC-30, ortholog of human PITX1-3, is required to induce VD terminal identity genes. Since UNC-30 is not expressed in cholinergic MNs (*Jin et al., 1994*), we hypothesized that UNC-3 prevents expression of UNC-30/PITX, leading to inactivation of VD terminal identity genes. However, this is not the case because: (1) ectopic *unc-30* expression is not observed in *unc-3*-depleted MNs, and (2) the ectopic expression of the VD marker (*ser-2*) in *unc-3* mutants was not abolished in *unc-3; unc-30* double mutants (*Figure 4—figure supplement 1*). These observations suggest that UNC-3 may act indirectly to prevent expression of VD and VC terminal identity genes through as yet unknown intermediary factors.

## The mid-body Hox protein LIN-39 (Scr/Dfd/Hox4-5) is the intermediary factor necessary for ectopic expression of VD and VC features in *unc-3* mutants

If the hypothesis of indirect repression is correct, mutation of the intermediary factor(s) in the *unc-3* mutant background would selectively eliminate ectopic expression of VD and/or VC terminal identity genes in *unc-3*-depleted MNs. To identify such factor(s), we embarked on an unbiased genetic screen. For the screen, we chose a transgenic *gfp* reporter strain for *flp-11*, an FMRF-like neuropeptide-encoding gene expressed in both VD and VC neurons (*Figure 1B*, *Figure 1—figure supplement 1*), which is markedly affected by UNC-3 (*Figure 4A–B*, *Figure 2—figure supplement 1*, panels D-E). We mutagenized *unc-3 (n3435); flp-11::gfp* animals with ethyl methanesulfonate (EMS) and visually screened ~4200 haploid genomes for mutants in which ectopic *flp-11::gfp* expression in

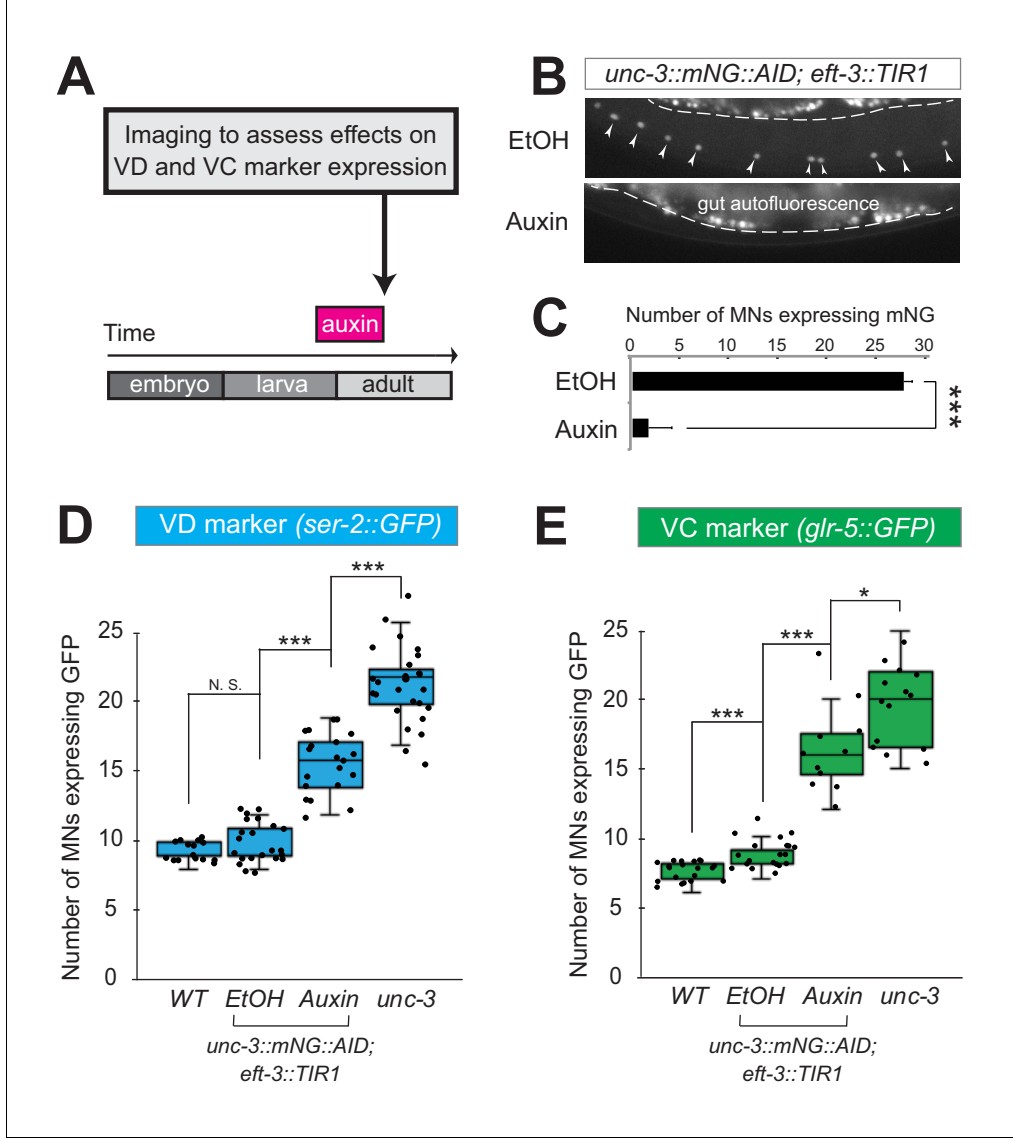

**Figure 3.** UNC-3 is continuously required to prevent expression of VD and VC terminal identity features. (**A**) Schematic showing time window of auxin administration. (**B**) Animals of the *unc-3::mNG::AID; eft-3::TIR1* genotype were either administered ethanol (EtOH) or auxin at the L4 stage. Twenty four hours later, expression of endogenous *unc-3* reporter (*unc-3::mNG::AID*) is severely reduced in the nuclei of VNC MNs (arrowheads) at the young adult stage (day 1). The same exact region was imaged in EtOH- and auxin-treated worms. mNG green fluorescent signal is shown in white for better contrast. White dotted line indicates the boundary of intestinal tissue (gut), which tends to be autofluorescent in the green channel. (**C**) Quantification of number of MNs expressing the *unc-3::mNG::AID* reporter after EtOH (control) and auxin treatment. N > 12. ***p<0.001. (**D**) Auxin or ethanol (control) were administered at larval stage 3 (L3) on *unc-3::mNG::AID; eft-3::TIR1* animals carrying the VD marker *ser-2::gfp.* Images were taken at the young adult stage (day 1.5). A significant increase in the number of MNs expressing the VD marker was evident in the auxin-treated animals compared to EtOH-treated controls. For comparison, quantification is provided for *ser-2::gfp* expressing MNs of wild-type animals and *unc-3(n3435)* mutants. Similar results were obtained when auxin was applied at L4 or day 1 adult animals. N > 20. ***p<0.001. (**E**) Auxin or ethanol (control) were administered at larval stage 4 (L4) on *unc-3::mNG::AID; eft-3::*TIR1 animals carrying the VC marker *glr-5::gfp.* Images were taken at the young adult stage (day 2). A significant increase in the number of MNs expressing the VC marker was evident in the auxin-treated animals compared to EtOH-treated controls. N > 11. *p<0.05; ***p<0.001.

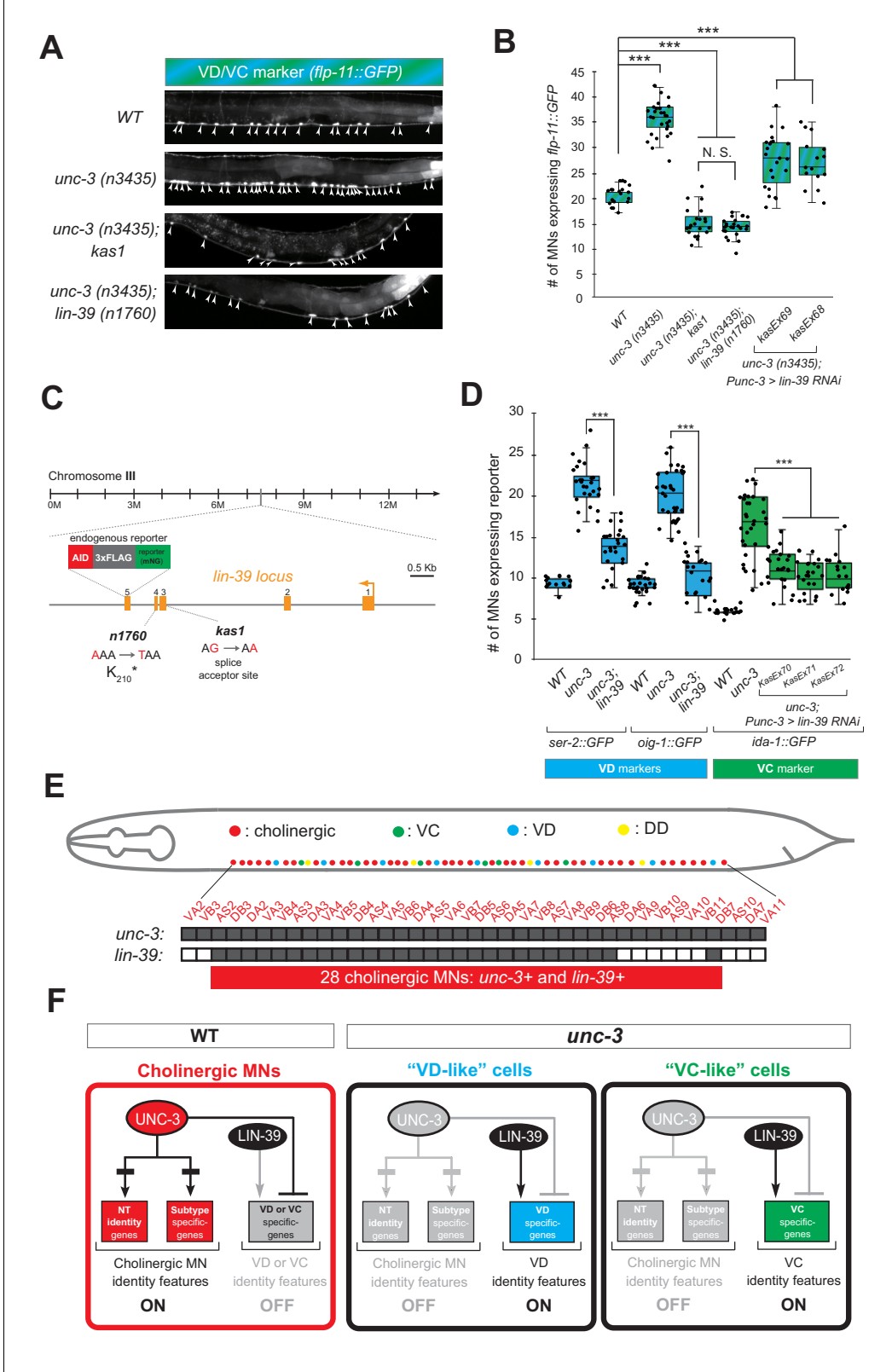

**Figure 4.** A genetic screen identifies the mid-body Hox protein LIN-39 (Scr/Dfd/Hox4-5) as necessary for ectopic expression of VD and VC terminal features. (**A**) Representative images of L4-stage WT, *unc-3(n3435)*, *unc-3(n3435); kas1*, and *unc-3(n3435); lin-39(n1760)* animals carrying *flp-11::gfp* (VD/VC marker). Arrowheads point to MN cell bodies with *gfp* marker expression. (**B**) Quantification graph summarizing results from panel A. The two right-most bars show quantification of two independent transgenic lines driving *lin-39 RNAi* specifically in cholinergic MNs (*Punc-3 >lin-39 RNAi*) of *unc-3*

Figure 4 continued

(n3435) mutants. N > 15. ***p<0.001. N.S: not significant. (C) Genetic locus of *lin-39*. Molecular lesions for *kas1* and *n1760* alleles are shown, as well as the *AID::3xFLAG::mNG* cassette inserted at the C-terminus (endogenous reporter). (D) Quantification of two VD (*ser-2::gfp*, *oig-1::gfp*) and one VC (*ida-1::gfp*) markers in WT, *unc-3 (n3435)*, *unc-3(n3435); lin-39(n1760)* animals at L4. The three right-most bars show quantification of three independent transgenic lines driving *lin-39* RNAi specifically in cholinergic MNs (*Punc-3 >lin-39 RNAi*) of *unc-3 (n3435)* mutants. N > 15. ***p<0.001. (E) Summary of *unc-3* and *lin-39* expression in cholinergic MNs. See *Figure 4—figure supplement 2* for raw data. (F) Schematic that summarizes our findings. In the wild type (*Faumont et al., 2011*) panel on the left, *lin-39* is normally expressed in cholinergic MNs but unable to induce expression of VD or VC genes. In the *unc-3* mutant, *lin-39* is now able to induce expression of alternative identity features (VD or VC) in distinct MN populations.

The online version of this article includes the following figure supplement(s) for figure 4:

**Figure supplement 1.** Ectopic expression of VD terminal identity markers in *unc-3* mutants requires LIN-39 but not UNC-30.

**Figure supplement 2.** LIN-39 is continuously required to activate distinct terminal identity genes in sex-shared and sex-specific cholinergic MNs.

*unc-3*-depleted MNs is suppressed. We isolated one mutant allele (*kas1*) (*Figure 4A–B*). The phenotype was 100% penetrant as all *unc-3 (n3435); flp-11::gfp* animals carrying *kas1* in homozygosity consistently displayed a dramatic reduction in ectopic *flp-11* expression.

Gross morphological examination of *unc-3 (n3435); kas1; flp-11::gfp* hermaphrodites revealed that, unlike *unc-3 (n3435); flp-11::gfp* animals, the introduction of *kas1* is accompanied by a lack of the vulva organ (vulvaless phenotype). Upon a literature survey for TF mutants that are vulvaless, we stumbled across the mid-body Hox gene *lin-39* (ortholog of Dfd/Scr in flies and Hox4-5 in vertebrates) (*Aboobaker and Blaxter, 2003*; *Clark et al., 1993*), and hypothesized that the molecular lesion of *kas1* may lie in the *lin-39* locus. Indeed, Sanger sequencing uncovered a point mutation on the splice acceptor site (WT: A<u>G</u> > *kas1*: A<u>A</u>) in the second intron of *lin-39* (*Figure 4C*). Similar to *unc-3 (n3435); kas1* animals, *unc-3 (n3435)* mutants carrying a previously published strong loss-of-function (premature STOP) allele of *lin-39 (n1760)* (*Clark et al., 1993*) displayed the same loss of ectopic *flp-11* expression (*Figure 4A–C*), suggesting that *kas1* is a loss-of-function mutation of *lin-39*. The ectopic expression of *flp-11* in *unc-3(n3435); kas1* animals can be, at least partially, rescued by (1) selective expression of *lin-39* cDNA in cholinergic MNs, and (2) introduction of the *lin-39* wild-type locus in the context of a ~ 30 kb genomic clone (fosmid) (*Figure 4—figure supplement 1*), corroborating that the *kas1* lesion in the *lin-39* locus is the phenotype-causing mutation.

Because *flp-11* is expressed in both VD and VC neurons, we next tested whether *lin-39* is required for ectopic expression of VD-specific (*ser-2*, *oig-1*) and VC-specific (*ida-1*) terminal identity genes in *unc-3*-depleted MNs. We found this to be the case by either generating *unc-3 (n3435); lin-39 (n1760)* double mutants (for VD markers) or by performing cholinergic MN-specific RNAi for *lin-39* in *unc-3 (n3435)* animals (for VC marker) (*Figure 4D*). RNAi was necessary because VC neurons do not survive in *lin-39 (n1760)* animals (*Potts et al., 2009*), and the use of the *n1760* allele could confound our VC marker quantifications. Of note, all other nerve cord MN subtypes are normally generated in *lin-39 (n1760)* single and *unc-3 (n3435); lin-39 (n1760)* double mutants (*Stefanakis et al., 2015*), indicating that suppression of the *unc-3* phenotype, that is, loss of ectopic VD gene expression in the double mutants is not due to MN elimination. Taken together, our genetic screen identified the mid-body Hox gene *lin-39* to be necessary for ectopic expression of both VD and VC terminal features in *unc-3*-depleted MNs (*Figure 4F*).

Interestingly, this finding contradicts our initial hypothesis of UNC-3 repressing an intermediary TF in order to prevent expression of VD and VC features because *lin-39* is co-expressed with (not repressed by) *unc-3* in wild-type cholinergic MNs at the mid-body region of the VNC (*Figure 4E*), as evident by our single-cell analysis of *unc-3* and *lin-39* reporters (*Figure 4—figure supplement 2*). Of note, 28 cholinergic MNs co-express *unc-3* and *lin-39*, which is in close agreement with the total number of VD-like (12.1 ± 2.6) and VC-like (10.5 ± 3.7) cells observed in *unc-3* mutants (*Figure 4E–F*). Moreover, we found that *lin-39* acts cell-autonomously as cholinergic MN-specific RNAi against *lin-39* in *unc-3 (n3435)* animals resulted in a significant reduction of ectopic terminal identity marker (*flp-11*, *ida-1*) expression (*Figure 4B–D*). In the following Results sections, we describe the molecular mechanism through which UNC-3 and LIN-39/Hox select and maintain throughout life key terminal features of cholinergic MNs (*Figure 4F*).

## UNC-3 prevents a switch in the transcriptional targets of LIN-39 in cholinergic motor neurons

What is the function of LIN-39 in wild-type cholinergic MNs of the VNC? Our previous findings suggested that LIN-39 and UNC-3, together with another mid-body Hox protein, MAB-5 (Antp/Hox6-8) (*Salser et al., 1993*), act synergistically to control expression of two cholinergic MN terminal identity genes (*unc-129,* ortholog of human BMP; *del-1*/Degenerin like sodium channel [ortholog of human SCNN1G]) (*Kratsios et al., 2017*). To test the extent of this synergy, we examined in *lin-39* and *mab-5* null animals the expression of 4 additional cholinergic MN terminal identity genes known to be controlled by UNC-3 (*acr-2*/nicotinic acetylcholine receptor; *dbl-1*/DPP/BMP-like; *unc-77*/sodium channel [ortholog of human NALCN], *slo-2*/potassium sodium-activated channel [ortholog of human KCNT1]) (*Kratsios et al., 2012*). In all four cases, we found a statistically significant decrease in *lin-39* mutants, and this effect was exacerbated in *lin-39; mab-5* double mutants (*Figure 5A–B*), indicating that the synergy of LIN-39 with MAB-5 (and UNC-3) extends to multiple terminal identity genes in cholinergic MNs (WT panel in *Figure 5D*). The observed effects were 100% penetrant and consistent with the previously described region-specific expression pattern of *lin-39* and *mab-5* in VNC MNs (*Figure 5A*) (*Kratsios et al., 2017*). Of note, while MAB-5 collaborates with LIN-39 to activate cholinergic MN identity genes (*Figure 5B*), it does not affect the ectopic expression of VD or VC genes observed in *unc-3* mutants (*Figure 5—figure supplement 1*).

Since UNC-3 controls directly, via its cognate binding site, cholinergic MN terminal identity genes (*Kratsios et al., 2012*), we then asked whether this is the case for LIN-39. We analyzed available ChIP-Seq data for LIN-39 from the modENCODE project (*Boyle et al., 2014*) and found evidence for direct LIN-39 binding in the *cis*-regulatory of all six cholinergic MN terminal identity genes (*unc-129, del-1, acr-2, dbl-1, unc-77, slo-2*) (*Figure 5C*, *Figure 5—figure supplement 1*). Moreover, we identified multiple consensus LIN-39 binding sites (previously defined as GATTGATG) (*Boyle et al., 2014*) located within the LIN-39 ChIP-Seq peaks in the *cis*-regulatory region of the aforementioned genes (*Supplementary file 2*).

This analysis strongly suggests that LIN-39, similar to UNC-3, regulates directly the expression of multiple terminal identity genes in cholinergic MNs (*Figure 5D*). However, in the absence of UNC-3, the function of LIN-39 in cholinergic MNs is modified. Instead of activating cholinergic MN identity genes, LIN-39 activates VD or VC terminal identity genes in *unc-3*-depleted MNs (*Figure 4*). Taken together, our data suggest that UNC-3 antagonizes the ability of LIN-39 to activate alternative identity genes, thereby preventing a switch in the transcriptional targets of LIN-39 (model schematized in *Figure 5D*). If this hypothesis is correct, one would expect decreased LIN-39 binding in the *cis*-regulatory region of cholinergic MN terminal identity genes in *unc-3* mutants. By performing ChIP-Seq for LIN-39 in *unc-3* mutant animals, we indeed observed decreased LIN-39 binding in the *cis*-regulatory region of the aforementioned genes (*Figure 5C*, *Figure 5—figure supplement 1*). As a positive control, LIN-39 binding in *unc-3* mutant animals was observed in other loci, including the *lin-39* locus itself (*Figure 5—figure supplement 1*), consistent with the known role of LIN-39 in regulating its own expression (*Niu et al., 2011*). Similar results were obtained by ChIP-qPCR for LIN-39 targets in *unc-3* mutants animals (*Figure 5—figure supplement 1*). We conclude that, in the absence of UNC-3, LIN-39 is released from cholinergic MN terminal identity gene promoters, presumably leading to increased availability of LIN-39 and thereby activation of alternative identity genes.

## LIN-39 is continuously required to control expression of terminal identity genes in cholinergic MNs

The neuronal function of Hox proteins at post-developmental stages is largely unknown (*Hutlet et al., 2016*). The continuous expression of mid-body Hox *lin-39* in both developing and adult cholinergic MNs led us to investigate whether *lin-39* is required to maintain expression of terminal identity genes in these neurons. To test this idea, we employed clustered regularly interspaced short palindromic repeats (CRISPR)/Cas9-based genome engineering and generated an auxin-inducible *lin-39* allele (*lin-39::mNG::3xFLAG::AID*) that also serves as an endogenous *lin-39* reporter (*mNG*). Animals carrying *lin-39::mNG::3xFLAG::AID* display no developmental phenotypes and show nuclear mNG expression in MNs located at the mid-body region of the VNC during development and adult stages (*Figure 6A*), corroborating previous observations with a LIN-39 antibody (*Maloof and Kenyon, 1998*). Upon crossing the *lin-39::mNG::3xFLAG::AID* animals with the *eft-3::*

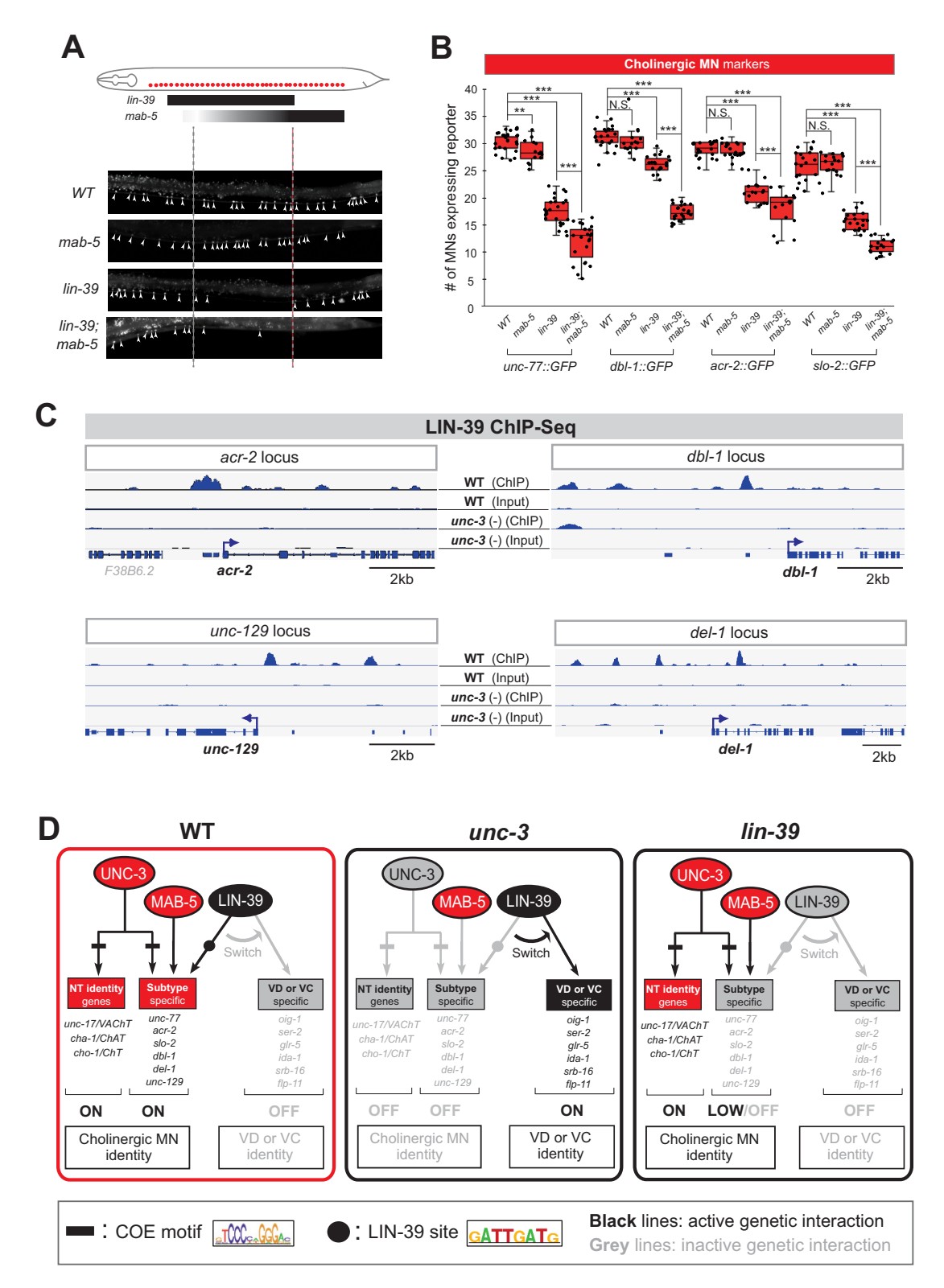

**Figure 5.** UNC-3 prevents a switch in the transcriptional targets of LIN-39 in cholinergic motor neurons. (**A**) Schematic summarizing the expression pattern of *lin-39* and *mab-5* in VNC cholinergic MNs. Below, representative images are shown of *unc-77::gfp* in *WT*, *lin-39 (n1760)*, *mab-5 (1239)* and *lin-39 (n1760); mab-5 (1239)* animals at L4 stage. Arrowheads point to MN cell bodies with *gfp* marker expression. Green fluorescence signal is shown in white for better contrast. Dotted black box indicates imaged area. (**B**) Quantification of cholinergic MN terminal identity markers (*unc-77, dbl-1, acr-2,*
*Figure 5 continued on next page*

*Figure 5 continued*

*slo-2*) in *WT, lin-39 (n1760), mab-5 (1239)* and *lin-39 (n1760); mab-5 (1239)* animals at L4. N > 15. **p<0.01; ***p<0.001. (C) ChIP-Seq tracks are shown for LIN-39 on four cholinergic MN terminal identity genes (*acr-2, unc-129, dbl-1, del-1*). The WT data come from the modENCODE project (*Boyle et al., 2014*), whereas the *unc-3 (-)* data were obtained by performing ChIP-Seq for LIN-39 on *unc-3 (n3435); lin-39 (kas9 [lin-39::mNG::3xFLAG::AID]* animals. (D) Schematic showing the transcriptional switch in LIN-39 targets. In WT animals, UNC-3, MAB-5 and LIN-39 co-activate subtype-specific genes in cholinergic MNs (e.g., *unc-77, dbl-1, unc-129, acr-2*). In *unc-3* mutants, LIN-39 is no longer able to activate these genes, and instead switches to VD- or VC-specific terminal identity genes. Black font: gene expressed. Gray font: gene not expressed. Gray arrows indicate inactive genetic interactions. COE motif taken from *Kratsios et al. (2012)* and LIN-39 site taken from *Weirauch et al. (2014)* are represented with black rectangles and dots, respectively. The online version of this article includes the following figure supplement(s) for figure 5:

**Figure supplement 1.** MAB-5 is not required for ectopic VD or VC marker expression and LIN-39 binding on cholinergic MN genes is affected in *unc-3* mutants.

---

*TIR1* line, we observed hypomorphic effects in the expression of two cholinergic MN identity genes (*acr-2, unc-77*) (*Figure 4—figure supplement 2*, panel C). Although LIN-39 protein is present in the nuclei of cholinergic MNs of *lin-39::mNG::3xFLAG::AID; eft-3::TIR1* animals (*Figure 6A*), these effects are likely due to a mild reduction in LIN-39 levels triggered by TIR1. However, post-embryonic auxin administration on these animals resulted in efficient LIN-39 protein depletion and significantly enhanced these effects (*Figure 6A–C*, *Figure 4—figure supplement 2*, panel C). We therefore conclude that LIN-39 is continuously required to maintain terminal identity features in cholinergic MNs.

Next, we sought to determine whether LIN-39 is continuously required for the ectopic activation of VD and VC terminal features observed in *unc-3* null animals. Indeed, auxin administration at L4 stage on *unc-3(n3435); lin-39::mNG::3xFLAG::AID* animals carrying either a VD (*ser-2*), VC (*glr-5*), or VD/VC (*flp-11*) marker resulted in a statistically significant suppression of the *unc-3* phenotype when compared to control (treated with ethanol) (*Figure 6D–F*).

To sum up, our findings with the auxin-inducible (*Figure 4—figure supplement 2*, panel C) and null *lin-39* alleles (*Figures 4F* and *5A–B*) indicate that, in the presence of UNC-3, LIN-39 is required to induce and maintain expression of cholinergic MN terminal identity genes (*Figure 5D*). In the absence of UNC-3 (*Figure 6*), LIN-39 is also continuously required - from development and possibly throughout life - for ectopic activation of VD and VC terminal identity genes (*Figure 5D*).

## LIN-39 is an activator of VD and VC terminal identity genes

The observation that *lin-39* is required for ectopic activation of both VD and VC terminal identity genes in *unc-3*-depleted MNs prompted us to examine the role of *lin-39* in VD and VC neurons of wild-type animals. Does LIN-39 control the same VD- and VC-specific terminal identity genes that become ectopically expressed in *unc-3* mutants?

To this end, we leveraged our endogenous *lin-39* reporter (*lin-39::mNG::3xFLAG::AID*) to assess expression in wild-type VD neurons at the mid-body region of the VNC, and found this to be the case (*Figure 7A*, *Figure 4—figure supplement 2*). Next, we found that LIN-39 is required to induce expression of VD terminal identity genes (*ser-2, oig-1*) (*Figure 7B*). To gain further mechanistic insights, we then asked whether *lin-39* acts together with UNC-30, the known activator of GABAergic MN identity genes (*Eastman et al., 1999*; *Jin et al., 1994*). Apart from confirming previous observations of UNC-30 controlling the VD-specific *oig-1* gene (*Cinar et al., 2005*; *Howell et al., 2015*), we also found that *ser-2* (*Figure 7B*) and *flp-11* (*Figure 4—figure supplement 2*, panel E) constitute novel UNC-30 targets in VD neurons. To test for synergistic effects, we focused on *ser-2* and *flp-11*, two VD-expressed terminal identity genes mildly affected in *lin-39* or *unc-30* single mutants. We generated *lin-39; unc-30* double mutants and observed stronger effects than either single mutant (*Figure 7B*, *Figure 4—figure supplement 2*, panel E). Such additive effects indicate that *lin-39* and *unc-30* act in parallel to activate VD terminal identity genes. Importantly, expression of other UNC-30 targets in GABAergic MNs, such as *flp-13* (DD-specific terminal identity marker) (*Cinar et al., 2005*; *Shan et al., 2005*; *Yu et al., 2017*) and genes expressed in both DD and VD neurons (*unc-25/GAD, unc-47/VGAT*), is unaffected in *lin-39* mutants (*Figure 7B*, *Figure 4—figure supplement 2*, panel F). Unlike UNC-30 that broadly controls multiple terminal features (NT identity and VD-specific terminal features) in VD neurons, we conclude that *lin-39* is selectively required for activation of VD-specific terminal identity genes (*Figure 7F*). To test for a maintenance role in VD

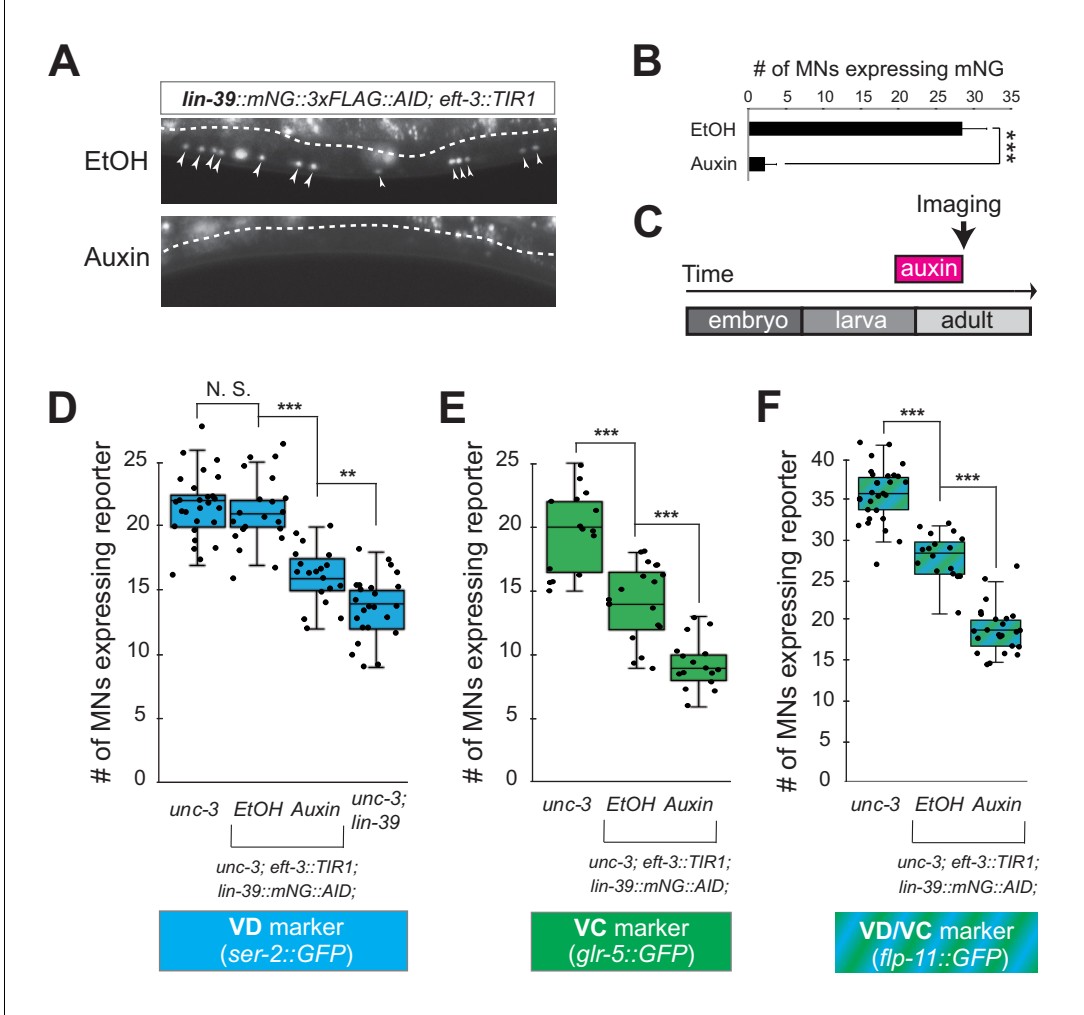

**Figure 6.** LIN-39 is continuously required to control expression of terminal identity genes. (**A**) Animals of the *lin-39::mNG::3xFLAG::AID; eft-3::TIR1* genotype were either administered ethanol (EtOH) or auxin at the L3 stage. Twenty four hours later, expression of endogenous *lin-39* reporter (*lin-39:: mNG::3xFLAG::AID*) is severely reduced in the nuclei of VNC MNs (arrowheads) at the young adult stage (day 1). mNG green fluorescent signal is shown in white for better contrast. White dotted line indicates the boundary of intestinal tissue (gut), which tends to be autofluorescent in the green channel. (**B**) Quantification of number of MNs expressing the *lin-39::mNG::3xFLAG::AID* reporter after EtOH (control) and auxin treatment. N > 14. ***p<0.001. (**C**) Schematic showing time window of auxin administration. (**D–F**) Auxin or ethanol (control) were administered at larval stage 4 (L4) on *unc-3 (n3435); lin-39::mNG::3xFLAG::AID; eft-3::TIR1* animals carrying either the VD marker *ser-2::gfp*, the VC marker *glr-5::gfp*, or the VD/VC marker *flp-11::gfp*. Images were taken at the young adult stage (day 1.6 for *ser-2*, day 1.8 for *glr-5* and day two for *flp-11*). A significant decrease in the number of MNs expressing the VD marker was evident in the auxin-treated animals compared to EtOH-treated controls. For comparison, quantification of marker expression is also provided in *unc-3 (n3435)* mutants. We note that hypomorphic effects in the ethanol treated group have been previously reported for other AID-tagged TFs in *C. elegans* (***Kerk et al., 2017***). Such effects appear to be target gene-specific, as they were observed for *glr-5* and *flp-11*, but not *ser-2* (***Figure 6E–F***). N > 15. **p<0.01, ***p<0.001, N. S: not significant.

neurons, we administered auxin at various post-developmental stages (L3, L4, day one adult) on animals carrying the *lin-39::mNG::3xFLAG::AID* allele. We found that LIN-39 is continuously required to maintain expression of the VD terminal identity gene *ser-2* (***Figure 7C***).

The above genetic analysis indicates that LIN-39 and UNC-30/PITX activate expression of VD-specific genes (left panel in ***Figure 7F***). Similarly, LIN-39 and UNC-3 directly co-activate terminal identity genes in cholinergic MNs (left panel in ***Figure 5D***). Since the absence of UNC-3 leads to ectopic activation of VD-specific genes (***Figure 5D***), we next considered the converse possibility: Does the absence of UNC-30/PITX lead to ectopic activation of cholinergic MN terminal identity genes in GABAergic VD neurons? However, this appears not to be the case as expression of 4 cholinergic

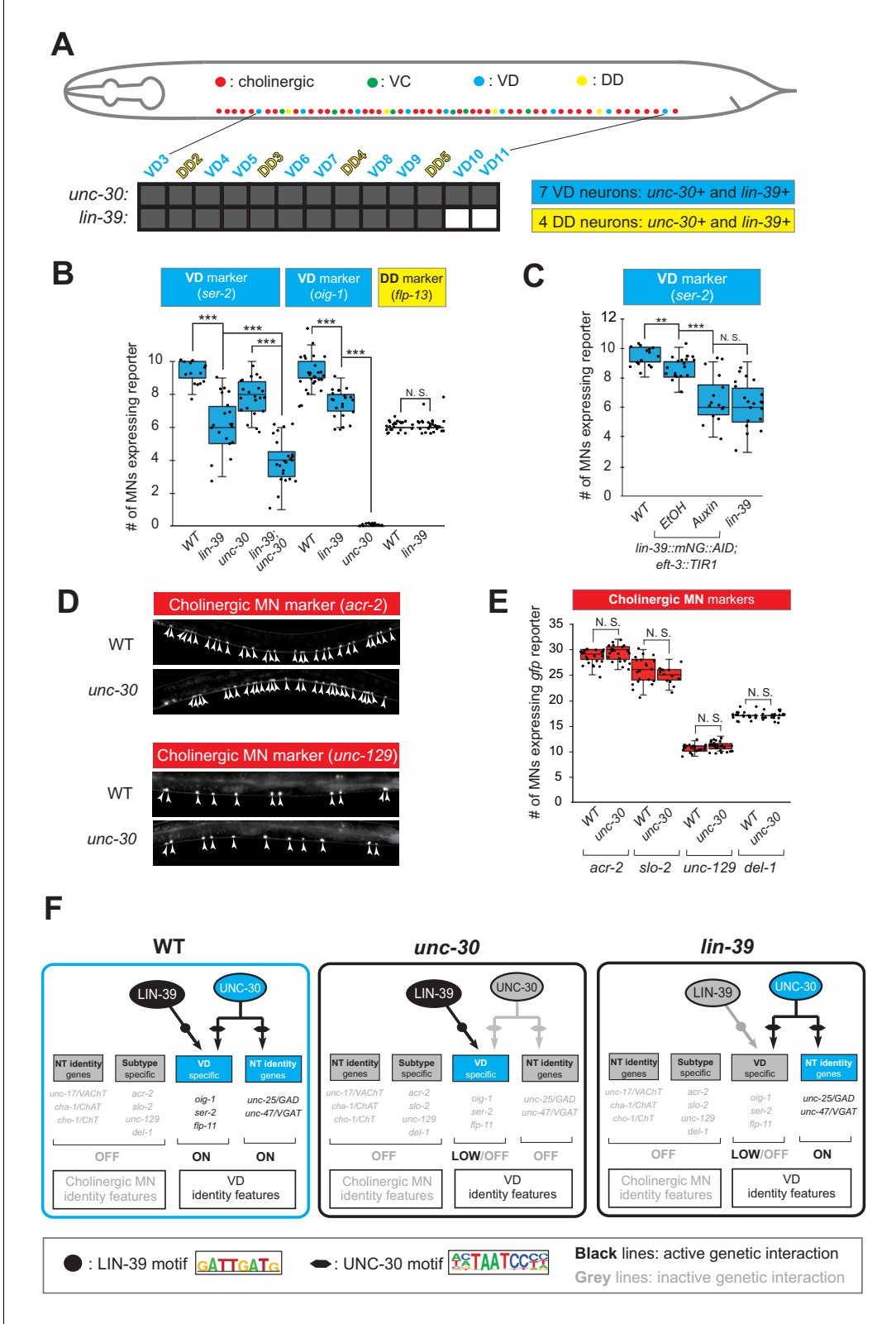

**Figure 7.** LIN-39 is an activator of VD terminal identity genes. (**A**) Schematic summarizing *unc-30* and *lin-39* expression in VD and DD neurons populating the VNC. In addition, 4 VD and 2 DD neurons are located in ganglia flanking the VNC (not shown because they were excluded from our analysis). Raw data on *lin-39* expression described in ***Figure 4—figure supplement 2***. (**B**) Quantification of two VD (*ser-2::gfp, oig-1::gfp*) and one DD (*flp-13::gfp*) markers in WT and *lin-39 (n1760)* animals at L4. Both VD markers were also tested in *unc-30 (e191)* mutants. Double *lin-39 (n1760); unc-30*

Figure 7 continued

*(e191)* mutants showed a more severe reduction in expression of the VD marker *ser-2::gfp* compared to each single mutant. N > 15. ***p<0.001. N. S: not significant. (C) Auxin or ethanol (control) were administered at larval stage 3 (L3) on *lin-39::mNG::3xFLAG::AID; eft-3::TIR1* animals carrying the VD marker *ser-2::gfp.* Images were taken at the young adult stage (day 1.5). A significant decrease in the number of MNs expressing the VD marker was evident in the auxin-treated animals compared to EtOH-treated controls. Similar results were obtained when auxin administration occurred at L4 or day one adult animals. For comparison, quantification of marker expression is also provided in WT and *lin-39 (n1760)* animals. N > 15. **p<0.01, ***p<0.001. N. S: not significant. (D) Several terminal identity markers of cholinergic neurons (*acr-2, slo-2, unc-129, del-1*) are not ectopically expressed in *unc-30-* depleted GABAergic MNs. A strong loss-of-function allele *e191* for *unc-30* was used (*Brenner, 1974*; *Eastman et al., 1999*). Arrowheads point to MN cell bodies with *gfp* marker expression. Green fluorescence signal is shown in white for better contrast. (E) Quantification of data presented in panel D. N. S: not significant. (F) Schematic summarizing the function of LIN-39 and UNC-30 in GABAergic VD neurons. LIN-39 site is taken from *Weirauch et al. (2014)*. UNC-30 site is taken from *Yu et al. (2017)*.

MN markers (*acr-2, slo-1, unc-129, del-1*), normally co-activated by UNC-3 and LIN-39 (*Figure 5A–B*), is unaffected in *unc-30* mutants (*Figure 7D–F*).

Similar to its role in sex-shared VD neurons, does *lin-39* control expression of terminal identity genes in sex-specific VC neurons? We used the auxin-inducible *lin-39::mNG::3xFLAG::AID* allele to address this question because, unlike all other nerve cord MNs, the VC neurons do not survive in *lin-39 (n1760)* null animals (*Potts et al., 2009*). We applied auxin at a late larval stage (L3-L4) to knock-down LIN-39 and observed that VC neurons do not die, providing an opportunity to test for putative effects on VC terminal identity gene expression. Indeed, we found a statistically significant reduction in the number of VC neurons expressing *srb-16* (compare auxin and ethanol in *Figure 4—figure supplement 2*, panel D).

Taken together, *lin-39* is required for expression of VD- and VC-specific terminal identity genes. In VD neurons, LIN-39 acts together with UNC-30/PITX to activate expression of VD-specific genes (left panel in *Figure 7F*). Collectively, these findings on VD and VC neurons together with observations on cholinergic MNs (*Figure 5D*) show that, in different MN subtypes, the mid-body Hox gene *lin-39* controls expression of distinct terminal identity genes, likely due to collaboration with distinct TFs (i.e., UNC-3 and MAB-5 in cholinergic MNs versus UNC-30 in VD neurons [compare *Figure 5D* and *Figure 7F*]).

## LIN-39 acts through distinct *cis*-regulatory elements to control *oig-1* expression in VD and VD-like motor neurons

Does LIN-39 act directly or indirectly to activate VD and VC terminal identity genes? Analysis of available ChIP-Seq data (modENCODE project) indicates direct regulation of these genes by LIN-39 (*Figure 8A*, *Figure 8—figure supplement 1*). However, the low resolution of ChIP-Seq data does not allow the identification of the exact DNA sequence recognized by LIN-39. Therefore, we interrogated the *cis*-regulatory region of two VD terminal identity genes *oig-1* and *ser-2* for the presence of the consensus LIN-39 binding site GATTGATG (*Boyle et al., 2014*) and found several copies located within the boundaries of the LIN-39 ChIP-Seq peaks in *oig-1* and *ser-2* (*Supplementary file 2*). To test the functionality of these putative LIN-39 binding sites, we honed in on *oig-1* and performed a systematic *cis*-regulatory analysis in the context of transgenic reporter animals. A previous study identified a minimal 125 bp *cis*-regulatory element (contained within the LIN-39 peak boundaries) upstream of *oig-1* as sufficient to drive reporter gene expression in VD neurons (*Howell et al., 2015*) (*Figure 8A*). We independently confirmed this observation, and further found that the 125 bp element contains a single LIN-39 site. Mutation of this site in the context of transgenic *oig-1* reporter animals (*oig-1* $^{125bp\ LIN-39\ site\ MUT}$*::tagRFP*) leads to a significant reduction of tagRFP expression in VD neurons (*Figure 8A–B*), phenocopying the effect observed in *lin-39 (n1760)* null mutants (*Figure 7B*). We conclude that, in wild-type animals, LIN-39 acts directly, by recognizing its cognate site, to activate expression of the VD-specific gene *oig-1*. Interestingly, a functional binding site for UNC-30/PITX also exists in this 125 bp element (*Howell et al., 2015*; *Yu et al., 2017*), and is spaced 11 base pairs apart from the LIN-39 site (*Figure 7D*), indicating that LIN-39 and UNC-30 control *oig-1* by recognizing distinct and in close proximity *cis*-regulatory motifs. Moreover, available UNC-30 ChIP-Seq data further support this possibility as UNC-30 and LIN-39 ChIP-Seq peaks largely overlap at this 125 bp element (*Figure 8A*). Lastly, deletion of the region where LIN-39 and UNC-30 peaks

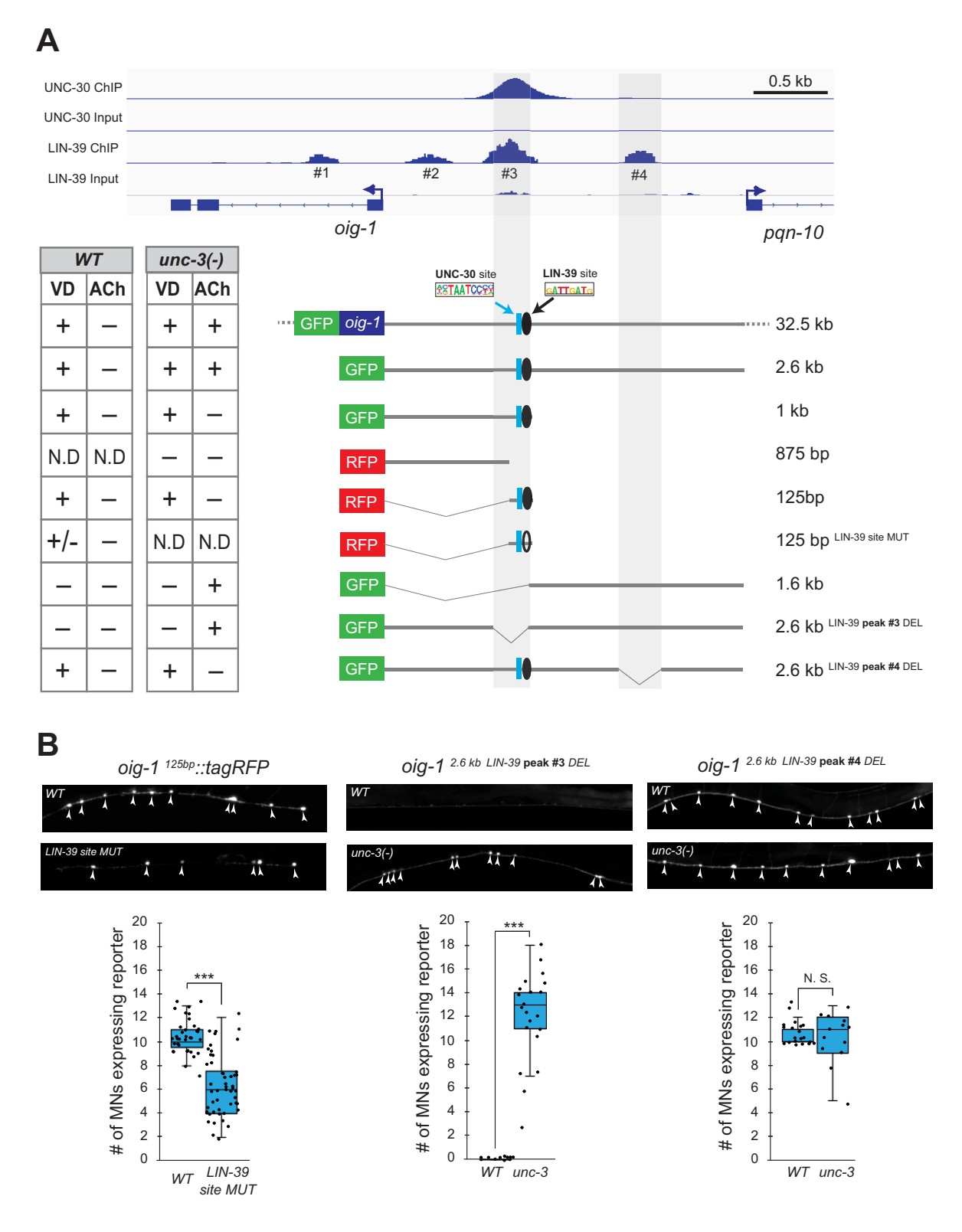

**Figure 8.** LIN-39 acts through distinct *cis*-regulatory elements to activate *oig-1* expression in VD and VD-like neurons. (**A**) ChIP-Seq tracks are shown for UNC-30 (the top two) and LIN-39 (the bottom two) on VD gene *oig-1* locus. The UNC-30 data were obtained from *Yu et al. (2017)*. The LIN-39 data come from the modENCODE project (*Boyle et al., 2014*). Four LIN-39 peaks are annotated with peak#3 largely overlapping with UNC-30 peak. The results of *cis*-regulatory analysis in both WT and *unc-3* mutants are shown in the lower panel (aligned to the ChIP-seq tracks). Expression patterns of at

Figure 8 continued

least two transgenic lines were analyzed for each construct. '+' indicates consistent and bright expression in ventral nerve cord (VNC) MNs (either VD or cholinergic). '+/−' indicates consistent and bright expression in noticeable less number of VNC MNs. '−' indicates no or extremely dim expression in VNC MNs. 'N.D.': Not determined. In the schematic of the transgenes, a known UNC-30 site is shown as a blue box and a bioinformatically predicted LIN-39 site is represented as a black circle (filled circle indicates the presence of the site while unfilled one indicates deletion of the site). MUT indicates deletion of the LIN-39 site and DEL indicates deletion of the respective LIN-39 peak region. (B) Images (top part) and quantifications (bottom part) of selected constructs in the *cis*-regulatory analysis shown in (A). Animals carrying the *oig-1$^{125bp}$::tagRFP* (left panel) with the LIN-39 site deleted show reduced tagRFP reporter expression in VD neurons; animals carrying the *oig-1$^{2.6kb\ LIN-39\ peak\ \#3\ DEL}$* (middle panel) ectopically express the reporter in cholinergic MNs of *unc-3* mutants, but not in WT animals; animals carrying the *oig-1$^{2.6kb\ LIN-39\ peak\ \#4\ DEL}$* (right panel) do not show ectopic reporter expression in *unc-3*-depleted MNs, but do show VD expression in both wild-type and *unc-3* mutants. N > 12. ***p<0.001. N. S: not significant. The online version of this article includes the following figure supplement(s) for figure 8:

**Figure supplement 1.** LIN-39 binds directly to the *cis*-regulatory region of VD and VC terminal identity genes.

overlap in the context of a 2.6 kb *oig-1* reporter (*oig-1 $^{2.6kb\ LIN-39\ peak\ \#3\ DEL}$*) abolish reporter expression in VD neurons (**Figure 8A–B**).

We next asked whether LIN-39 acts through the same or distinct *cis*-regulatory elements to drive *oig-1* expression in VD versus VD-like neurons of *unc-3* mutants. While *oig-1* reporters in the context of a large (32.5 kb) genomic clone (fosmid) or a 2.6 kb intergenic region do show expression in both VD and VD-like neurons of *unc-3* mutants, reporter animals carrying 1 kb of *cis*-regulatory sequence (that contains the 125 bp element) immediately upstream of ATG showed expression only in VD neurons (**Figure 8A**). Conversely, a distal 1.6 kb element displayed expression in VD-like cells, but no expression in VD neurons of either WT or *unc-3* animals, suggesting the VD and VD-like elements are physically separated on the genome. Within the 1.6 kb element, there is a LIN-39 binding peak (peak #4) based on available ChIP-Seq data on WT animals. Deletion of this peak in the context of a 2.6 kb *oig-1* reporter (*oig-1 $^{2.6kb\ LIN-39\ peak\ \#4\ DEL}$*) resulted in loss of expression in VD-like cells of *unc-3* mutants, whereas reporter expression was maintained in VD neurons (**Figure 8A–B**). This analysis strongly suggests that LIN-39 acts through distinct *cis*-regulatory elements to activate *oig-1* expression in VD versus VD-like cells.

## The LIN-39-mediated transcriptional switch depends on UNC-3 and LIN-39 levels

How does the absence of UNC-3 lead to ectopic and *lin-39*-dependent activation of VD terminal identity genes in cholinergic MNs (**Figure 9D**)? In principle, UNC-3 and LIN-39 could physically interact in order to co-activate expression of cholinergic MN terminal identity genes. In the absence of *unc-3*, this interaction would be disrupted and LIN-39 becomes available, in cholinergic MNs, to assume its VD function, that is to activate VD-specific terminal identity genes (**Figure 9D**). Although our co-immunoprecipitation (co-IP) experiments on UNC-3 and LIN-39 in a heterologous system (HEK cells) did not provide evidence for physical interaction (**Figure 9—figure supplement 1**), the heterologous context of this experiment still leaves open the possibility that, in cholinergic MNs in vivo, UNC-3 directly (or indirectly) recruits LIN-39 on terminal identity gene promoters. This scenario is supported by the observed decrease of LIN-39 binding on cholinergic MN gene loci in *unc-3* mutants (**Figure 5C**). Lastly, the gene dosage experiments presented below firmly suggest there is a close stoichiometric relationship between UNC-3 and LIN-39, reminiscent of LIM homeodomain TF stoichiometries described in vertebrate MNs (**Song et al., 2009**).

Because the decrease of LIN-39 binding is accompanied by ectopic activation of VD terminal identity genes in *unc-3*-depleted MNs, we hypothesized that LIN-39 is the rate-limiting factor present in limited amount in cholinergic MNs. That is, in the presence of UNC-3, LIN-39 activates cholinergic MN identity genes, but in its absence LIN-39 becomes available to activate alternative identity (e.g., VD) genes. Quantification of the endogenous expression levels of both proteins indeed showed lower levels of LIN-39 expression compared to UNC-3 (**Figure 9A**). Supporting the aforementioned hypothesis, we found a gene dosage relationship between *unc-3* and *lin-39*. Loss of one *unc-3* copy (*unc-3 (n3435)/+*) caused slight ectopic expression of VD genes (*ser-2* in **Figure 9B** and *flp-11* in **Figure 9—figure supplement 2**, panel A), but that ectopic expression is decreased by loss of one *lin-39* copy (*unc-3 (n3435)/+; lin-39 (n1760)/+*) (**Figure 9B**, **Figure 9—figure supplement 2**, panel A). Accordingly, loss of one *lin-39* copy in *unc-3* null animals (*unc-3 (n3435); lin-39 (n1760)/+*)

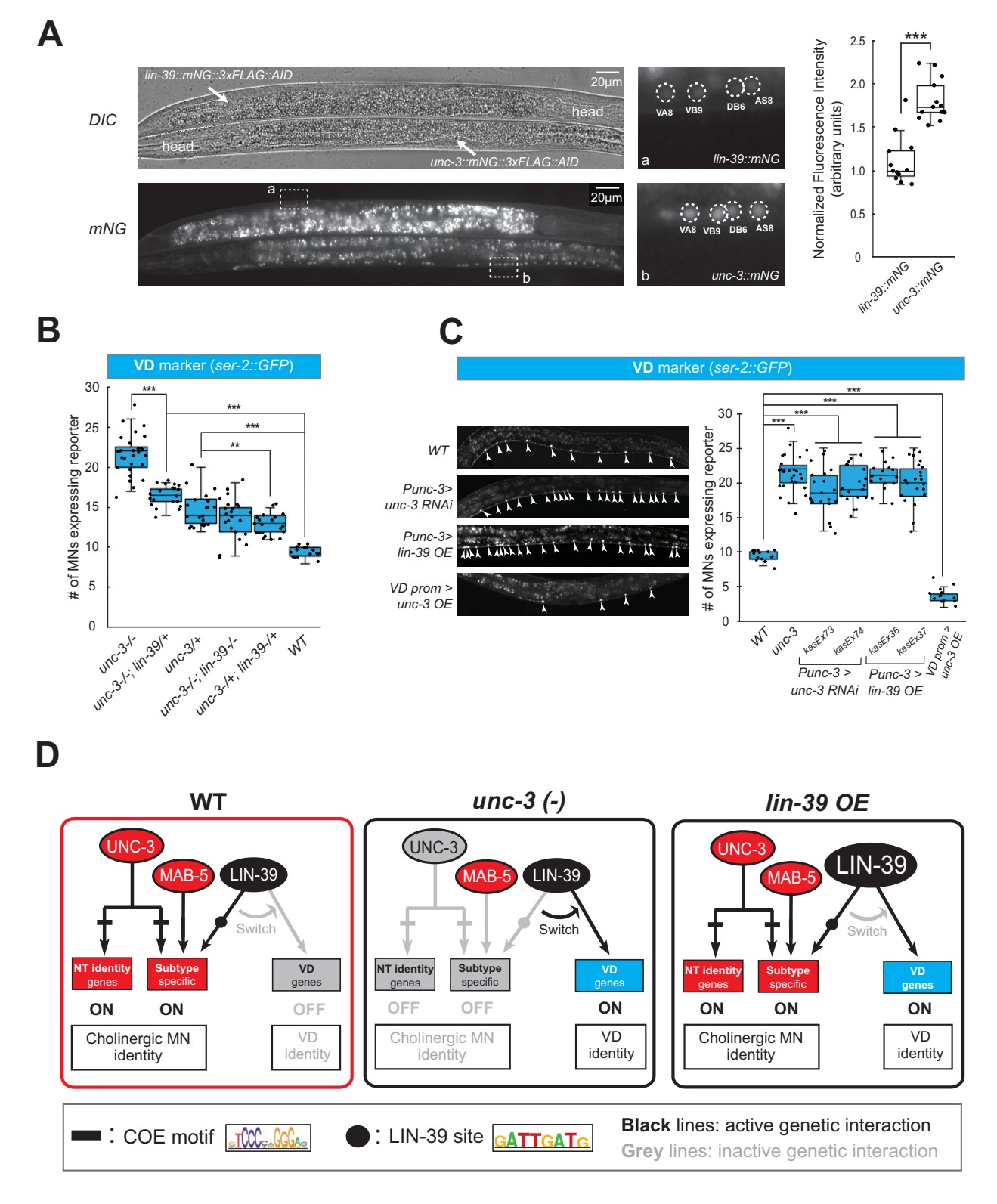

**Figure 9.** Gene dosage experiments suggest that LIN-39 is the rate-limiting factor. (**A**) The endogenous expression levels of UNC-3 are higher than LIN-39. The DIC and mNG channels of two worms next to each other on the same slide with the genotype of *lin-39::mNG::3xFLAG::AID* (top, anterior right, ventral up) and *unc-3::mNG::3xFLAG::AID* (bottom, anterior left, ventral down) respectively are shown on the left. VNC regions indicated by dashed frame (a, b) are zoomed in at the middle panel with dashed circles around MN nuclei. The identities of these MNs are shown (e.g., VA8, VB9).
*Figure 9 continued on next page*

*Figure 9 continued*

The same cholinergic MNs have stronger expression levels of endogenous *unc-3::mNG* than *lin-39::mNG*. Quantification of the fluorescence intensities is shown on the right panel. For details on the quantification, see Materials and Materials and methods. N = 12. ***p<0.001. (**B**) Quantification of the VD marker (*ser-2::gfp*) in *unc-3 (n3435)*, *unc-3 (n3435); lin-39 (n1760)/+*, *unc-3 (n3435)/+*, *unc-3 (n3435); lin-39 (n1760)*, *unc-3 (n3435)/+; lin-39 (n1760)/+*, and WT animals at L4. N > 15. **p<0.01, ***p<0.001. (**C**) Representative images of the VD marker (*ser-2::gfp*) expression on the left in L4 stage transgenic animals that either down-regulate *unc-3* in cholinergic MNs (*Punc-3 >unc-3 RNAi*), over-express *lin-39* in cholinergic MNs (*Punc-3 >lin-39 OE*), or over-express *unc-3* in VD neurons (VD prom [*unc-47* prom]>*unc-3 OE*). Arrowheads point to MN cell bodies with *gfp* marker expression. Green fluorescence signal is shown in white for better contrast. Quantification is provided on the right. Two independent transgenic lines were used for *Punc-3 >unc-3 RNAi* and *Punc-3 >lin-39 OE*. N > 13. ***p<0.001. (**D**) Schematic summarizing the gene dosage experiments.

The online version of this article includes the following figure supplement(s) for figure 9:

**Figure supplement 1.** UNC-3 does not physically interact with LIN-39 in a heterologous system.

**Figure supplement 2.** UNC-3 and LIN-39 levels are crucial for ectopic expression of VD/VC terminal identity marker *flp-11*.

also reduced, but did not eliminate, ectopic expression of VD genes (*Figure 9A*, *Figure 9—figure supplement 2*, panel A). Moreover, knock-down of *unc-3* with RNAi specifically in cholinergic MNs also led to ectopic expression of VD genes (*Figure 9C*, *Figure 9—figure supplement 2*, panels B-C), whereas ectopic expression of UNC-3 in VD neurons resulted in repression of VD gene expression, presumably by recruiting LIN-39 away from VD promoters (*Figure 9C*, *Figure 9—figure supplement 2*, panels B-C). Lastly, we asked whether LIN-39 is sufficient to induce expression of VD terminal identity genes in cholinergic MNs. Indeed, we found this to be the case (*Figure 9C*, *Figure 9—figure supplement 2*, panels D-E). In conclusion, we propose that the LIN-39-mediated transcriptional switch observed in *unc-3* mutants critically depends on UNC-3 and LIN-39 levels, with the latter being the rate-limiting factor (*Figure 9D*).

## Ectopic expression of VD terminal identity genes in cholinergic motor neurons is associated with locomotion defects

The dual role of UNC-3 revealed by our molecular analysis (*Figure 2*) led us to posit that the severe locomotion defects observed in *unc-3* animals may represent a composite phenotype (*Brenner, 1974*; *Yemini et al., 2013*). In other words, these defects are not only due to loss of expression of cholinergic MN terminal identity determinants (e.g., *unc-17*/VAChT, *cha-1*/ChAT, *del-1*/Degenerin-like sodium channel, *acr-2*/acetylcholine receptor [ortholog of CHRNE]), but also due to the ectopic expression of VD and VC terminal features (e. g., *ser-2*/serotonin receptor [ortholog of HTR1D], *flp-11*/FRMR-like neuropeptide, *glr-5*/Glutamate receptor [ortholog of GRID], *srb-16*/GPCR) in *unc-3*-depleted MNs. To genetically separate these distinct molecular events, we generated *unc-3 (n3435); lin-39 (n1760)* double mutants, which do display loss of cholinergic MN terminal identity genes, but the ectopic expression of VD and VC terminal features is suppressed (*Figure 4F*). We predicted that if ectopic expression of VD and VC genes contributes to locomotion defects, then *unc-3 (n3435)* mutants would display more severe locomotion defects than *unc-3 (n3435); lin-39 (n1760)* double mutants. To test this, we performed high-resolution behavioral analysis of freely moving adult (day 1) *C. elegans* animals using automated multi-worm tracking technology (*Javer et al., 2018b*; *Yemini et al., 2013*). This analysis can quantitate multiple features related to *C. elegans* locomotion (*e.g.*, speed, crawling amplitude, curvature, pause, forward and backward locomotion) and, most importantly, each feature can be localized to a specific part of the nematode's body (e. g., head, mid-body, tail). Since *unc-3* and *lin-39* expression uniquely overlaps in mid-body nerve cord MNs that innervate mid-body muscles, we hypothesized that loss of *unc-3* and/or *lin-39* genes would have effects on mid-body posture and motion, and thereby focused our analysis on mid-body curvature features. Of the 49 mid-body features examined, 29 were significantly different in *unc-3* single mutants when compared to wild-type (N2 strain) animals (see *Supplementary file 3* for all 49 features). Intriguingly, 12 of these 29 features (41.37%) were significantly suppressed in *unc-3; lin-39* double mutants (*Figure 10A*, *Figure 10—figure supplement 1*, panel A, *Supplementary file 3*), suggesting that suppression of these behavioral defects could be attributed to suppression of the ectopically expressed VD and VC terminal identity genes in these double mutants. We found no evidence for suppression of the remaining 17 features in *unc-3; lin-39* double mutants, likely due to the fact that UNC-3 also controls other terminal identity genes, such as NT pathway genes (*Figure 9D*), independently of LIN-39.

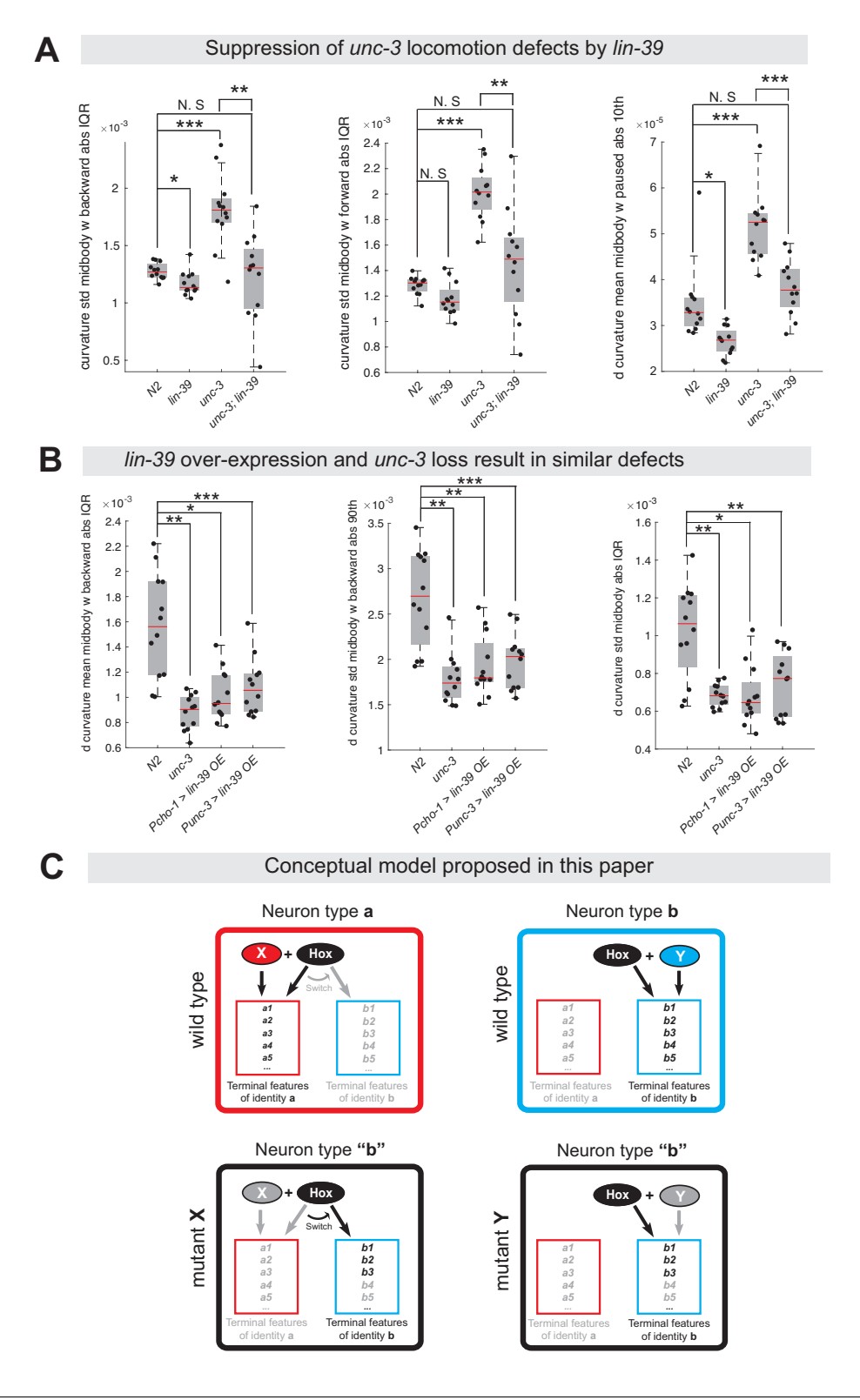

**Figure 10.** Ectopic expression of VD terminal identity genes in cholinergic motor neurons is associated with locomotion defects. (**A**) Examples of three mid-body locomotion features that are significantly affected in *unc-3 (n3435)* animals, but markedly improved in *unc-3 (n3435); lin-39 (n1760)* double mutant animals. Each black dot represents a single adult animal. The unit for the first two graphs is 1/microns. The unit for the graph on the right is 1/(microns*seconds). N = 12. Additional mid-body features affected in *unc-3 (n3435)* animals, but improved in *unc-3 (n3435); lin-39 (n1760)* mutants are

Figure 10 continued

provided in *Figure 10—figure supplement 1* and *Supplementary file 3*. *p<0.01, **p<0.001, ***p<0.0001. (B) Examples of three mid-body locomotion features affected in *unc-3 (n3435)* mutants and animals over-expressing *lin-39* in cholinergic MNs. Each black dot represents a single adult animal. The unit for the Y axis is 1/(microns*seconds). N = 12. Additional mid-body features affected in *unc-3 (n3435)* and *lin-39* over-expressing animals are provided in *Figure 10—figure supplement 1* and *Supplementary file 3*. *p<0.01, **p<0.001, ***p<0.0001. (C) Conceptual model summarizing the findings of this paper. Gray font: not expressed gene. Black font: expressed gene. Gray arrow: inactive genetic interaction. Black arrow: active genetic interaction.

The online version of this article includes the following figure supplement(s) for figure 10:

**Figure supplement 1.** Automated worm tracking analysis on *unc-3* and *unc-3; lin-39* mutants.

Next, we asked whether ectopic expression of VD terminal identity genes in otherwise wild-type animals can lead to locomotion defects. To test this, we took advantage of our transgenic animals that selectively over-express LIN-39 in cholinergic MNs (*Pcho-1 >LIN-39*, *Punc-3 >LIN-39*) (*Figure 9C*, *Figure 10—figure supplement 1*, panel D). First, we confirmed that in these animals expression of cholinergic MN terminal identity genes is unaffected (*Figure 10—figure supplement 1*, panel D). Second, we found that LIN-39 overexpression led to ectopic activation of VD, but not VC, terminal identity genes in cholinergic MNs (*Figure 10—figure supplement 1*, panel D), providing an opportunity to specifically assess the consequences of ectopic VD gene expression on animal locomotion. We found that 9 of the 29 (31.03%) mid-body features affected in *unc-3 (n3435)* animals were also altered in animals over-expressing *lin-39* in cholinergic MNs (*Figure 10B*, *Figure 10—figure supplement 1*, panels B-C, *Supplementary file 3*).

In conclusion, our behavioral analysis is in agreement with our molecular findings. At the molecular level, we found that *lin-39*/Hox is necessary for the ectopic expression of VD terminal identity genes in *unc-3* mutants. At the behavioral level, this *lin-39*-dependent, ectopic expression of terminal identity genes is accompanied by locomotion defects.

## Discussion

During development, individual neuron types must select their unique terminal identity features, such as expression of NT receptors, ion channels and neuropeptides. Continuous expression of these features - from development through adulthood - is essential for safeguarding neuronal terminal identity, and thereby ensuring neuronal function (*Deneris and Hobert, 2014*; *Hobert, 2011*; *Hobert, 2016*). Here, we provide critical insights into the mechanisms underlying selection and maintenance of neuron type-specific terminal identity features by using the well-defined MN populations of the *C. elegans* nerve cord as a model. First, we report that, in cholinergic MNs, the terminal selector-type TF UNC-3 has a dual role; UNC-3 is not only required to promote cholinergic MN identity features (*Kratsios et al., 2012*), but also to prevent expression of multiple terminal features normally reserved for three other ventral cord neuron types (VD, VC, CA). Second, we provide evidence that cholinergic MNs can secure their terminal identity throughout life by continuously relying on UNC-3's dual function. Third, we propose an unusual mechanism underlying this dual function, as we find UNC-3 necessary to prevent a switch in the transcriptional targets of the mid-body Hox protein LIN-39 (Scr/Dfd/Hox4-5) (*Figure 9D*). Lastly, our findings shed light upon the poorly explored, post-embryonic role of Hox proteins in the nervous system by uncovering that LIN-39 is continuously required to maintain expression of multiple terminal identity genes in MNs.

### UNC-3 determines the function of the rate-limiting factor LIN-39/Hox in cholinergic motor neurons

Numerous cases of neuron type-specific TFs with a dual role have been previously described in both vertebrate and invertebrate models systems (*Britanova et al., 2008*; *Cheng et al., 2004*; *Kala et al., 2009*; *Lopes et al., 2012*; *Mears et al., 2001*; *Morey et al., 2008*; *Nakatani et al., 2007*; *Sagasti et al., 1999*). Although the underlying mechanisms often remain unclear, recent studies proposed two modes of action. First, such TFs can act directly to activate 'desired' terminal identity features and repress (also directly) alternative identity features (*Lodato et al., 2014*; *Wyler et al., 2016*). Second, neuron type-specific TFs can act indirectly by controlling intermediary factors. For example, in the mouse spinal cord, a complex of three TFs (Isl1, Lhx3, NLI) specifies MN

identity by recognizing specific DNA elements in the *cis*-regulatory region of MN-specific genes and the homeodomain TF Hb9, and activates their expression. Hb9 functions as a transcriptional repressor of alternative (V2a interneuron) identity genes, thereby consolidating MN identity (*Lee et al., 2008*; *Song et al., 2009*; *Thaler et al., 2002*). An analogous mechanism operates in V2a interneurons and involves Chx10, a homeodomain protein that represses alternative neuronal identity programs (*Clovis et al., 2016*; *Lee et al., 2008*). Hence, mouse mutants for *Hb9* or *Chx10* result in ectopic expression of alternative identity genes (*Arber et al., 1999*; *Clovis et al., 2016*; *Thaler et al., 1999*). Several TFs (*unc-4*/Uncx, *mab-9*/Tbx20, *unc-55*/COUP, *bnc-1*/BNC) with repressor activity are known to control aspects of cholinergic MN development in *C. elegans* (*Kerk et al., 2017*; *Pflugrad et al., 1997*; *Pocock et al., 2008*; *Von Stetina et al., 2007*; *Winnier et al., 1999*). However, their genetic removal did not result in ectopic expression of alternative (VD, VC) identity features in cholinergic MNs (data not shown). Although we cannot exclude the involvement of yet-to-be identified transcriptional repressors acting downstream of UNC-3, our genetic and biochemical analyses led us to propose the following mechanism underlying UNC-3's dual role.

In cholinergic MNs, *unc-3* and *lin-39* are co-expressed, albeit the latter in lower levels (*Figure 9A*), suggesting that LIN-39/Hox is a rate-limiting factor whose function is determined by UNC-3. In wild-type animals, UNC-3 and LIN-39 occupy *cis*-regulatory elements of cholinergic MN terminal identity genes, resulting in their activation (*Figure 5C*) (*Kratsios et al., 2012*). In the absence of UNC-3, LIN-39 is released from these elements and becomes available to activate alternative identity genes, such as VD-specific terminal identity genes. Several lines of evidence support this conclusion. First, ChIP-seq data show that LIN-39 binding is decreased in cholinergic MN gene loci in *unc-3* mutants (*Figure 5C*). Second, our gene dosage experiments show that either lowering *unc-3* levels or increasing *lin-39* levels in cholinergic MNs results in ectopic activation of VD identity genes (*Figure 9B–C*). Lastly, we performed an extensive *cis*-regulatory analysis of one VD-specific gene (*oig-1*) and identified the element through which LIN-39 acts to induce *oig-1* expression in VD-like cells of *unc-3* mutants (*Figure 8*). Together, these data suggest that the role of UNC-3 in cholinergic MNs is not simply to activate gene expression with LIN-39, but also to 'recruit' LIN-39 away from promoters of alternative identity genes, thereby antagonizing its ability to activate those genes. Supporting this scenario, ectopic UNC-3 expression in VD neurons results in decreased expression of VD-specific genes (*Figure 9C*). Given that the mouse ortholog of *unc-3*, Ebf2, is co-expressed with Hox genes in cholinergic MNs of the spinal cord (*Catela et al., 2019*; *Kratsios et al., 2017*), the molecular mechanism described here may be conserved across species. Interestingly, a seminal study recently described a conceptually similar mechanism in the mouse retina, where CRX recruits MEF2D to retina-specific enhancers, resulting in selective activation of photoreceptor genes (*Andzelm et al., 2015*).

## Insights into how neurons maintain their terminal identity features throughout life

Is there a need for mechanisms that continuously prevent expression of alternative identity features in a post-mitotic neuron? Or, do such mechanisms become superfluous once neurons have restricted their developmental potential by committing to a specific terminal identity? This fundamental question is poorly explored, in part due to the fact that most neuron type-specific TFs have been studied during embryonic stages. For example, it is not known whether CRX is continuously required to activate retina-specific enhancers and simultaneously prevent expression of alternative identity genes (*Andzelm et al., 2015*). Our temporally controlled protein depletion experiments uncovered a continuous requirement for the dual role of UNC-3. Post-embryonic depletion of UNC-3 not only results in failure to maintain cholinergic MN terminal features (*Kratsios et al., 2012*), but is also accompanied by ectopic expression of alternative identity features (e.g., VD, VC). These findings reveal a simple and economical mechanism that can enable individual neuron types to select and maintain their distinct terminal identity features. That is, the same TF is continuously required - from development throughout life - to not only activate neuron type-specific identity genes, but also prevent expression of alternative identity features.

## Maintenance of terminal identity features: A new function of hox proteins in the nervous system

Across model systems, a large body of work on motor neurons and other neuron types has established that, during early development, Hox proteins are required for neuronal diversity, cell survival, axonal path finding and circuit assembly (*Baek et al., 2013*; *Catela et al., 2016*; *Estacio-Gómez and Díaz-Benjumea, 2014*; *Estacio-Gómez et al., 2013*; *Karlsson et al., 2010*; *Mendelsohn et al., 2017*; *Miguel-Aliaga and Thor, 2004*; *Moris-Sanz et al., 2015*; *Philippidou and Dasen, 2013*). However, the function and downstream targets of Hox proteins during post-embryonic stages are largely unknown. Our contributions towards this knowledge gap are twofold. First, we found that the mid-body Hox protein LIN-39 is continuously required, from development through adulthood, to control expression of MN terminal identity genes, thereby revealing a novel role for Hox proteins in maintaining neuronal identity. Second, we uncovered multiple terminal identity genes as downstream targets of LIN-39 in different MN subtypes (cholinergic MNs: *acr-2, dbl-1, unc-77, slo-2*; VD neurons: *oig-1, ser-2, flp-11*; VC neurons: *srb-16)*. Since continuous expression of these genes is essential for MN function, these findings may provide a molecular explanation for the uncoordinated locomotion defects observed in *lin-39* mutants (*Figure 10—figure supplement 1*). Given the maintained expression of Hox genes in the adult nervous system of flies, mice and humans (*Baek et al., 2013*; *Takahashi et al., 2004*; *Hutlet et al., 2016*), our findings may be broadly transferable.

## Impact on the concept of terminal selector genes

TFs able to broadly activate many distinct terminal identity features of a specific neuron type (e.g., NT biosynthesis components, NT receptors, ion channels, neuropeptides) have been termed 'terminal selectors' (*Hobert, 2008*). Several dozens of terminal selectors have been described thus far in multiple model systems including worms, flies and mice (*Hobert, 2011*; *Hobert, 2016*; *Hobert and Kratsios, 2019*). However, it is unclear whether terminal selectors are also required to prevent expression of alternative identity features. Our findings suggest this to be the case by revealing a dual role for UNC-3, the terminal selector of cholinergic MN identity in *C. elegans*. In the future, it will be interesting to see whether other terminal selectors also exert a dual role in order to safeguard neuronal terminal identity. Supporting this possibility, Pet-1, the terminal selector of mouse serotonergic neurons has been recently shown to repress several terminal identity genes (*Wyler et al., 2016*).

## Limitations and lessons learned about the control of neuronal terminal identity

The examination of multiple MN terminal identity markers at single-cell resolution enabled us to make an interesting observation. Although all *unc-3*-depleted nerve cord MNs uniformly lose their cholinergic identity, one subpopulation acquires VD terminal features ('VD-like' neurons) and another subpopulation acquires VC terminal features ('VC-like' neurons). This intriguing observation may be analogous to findings described in the mammalian neocortex, where genetic removal of the TF *Satb2* leads to loss of pyramidal neuron identity (UL1 subtype), and concomitant gain of molecular features specific to two other pyramidal neuron subtypes (DL, UL2) (*Britanova et al., 2008*). Together, the cases of UNC-3 and Satb2 support the notion that neuron type-specific TFs often suppress features of functionally related neuronal subtypes (*Arlotta and Hobert, 2015*).

Although our study employs an extensive repertoire of terminal identity markers for distinct MN subtypes, the extent of alternative identity features (e.g., VD, VC) being ectopically expressed in *unc-3*-depleted MNs remains unknown. Future unbiased transcriptional profiling of *unc-3*-depleted MNs could help address this issue. In addition, the strong axonal defects in MNs of *unc-3* mutants preclude any further attempts to assess whether the observed VD-like and VC-like cells, as defined by molecular markers, also acquire morphological features of VD and VC neurons, respectively (*Prasad et al., 1998*). However, the VD-like neurons of *unc-3* mutants do not acquire GABAergic identity like wild-type VD neurons (*Figure 2—figure supplement 1*, panel A), arguing against a complete cell fate transformation.

## Evolutionary implications of this study

Our findings highlight the employment of economical solutions to evolve novel cell types in the nervous system. The same Hox protein (LIN-39) collaborates with distinct terminal selectors in different MNs, and this collaboration determines the specificity of LIN-39/Hox function. In GABAergic (VD) neurons, LIN-39 works together with UNC-30/PITX to control expression of VD terminal identity genes, whereas in cholinergic MNs LIN-39 synergizes with UNC-3 to control cholinergic MN identity genes (*Figure 7F*, *9D*). We speculate that the *unc-3* mutant 'state' may constitute the 'ground state'. That is, the 'VD-like' neurons, for example, in *unc-3* mutants that express LIN-39 may represent an ancient cell type that was altered to become a new cell type through the recruitment of distinct terminal selectors (conceptual model in *Figure 10C*). Hence, the amount of genetic information required for evolution of new cell types is kept to minimum. The recruitment of UNC-30/PITX enabled 'VD-like' cells to fully adopt GABAergic VD neuron terminal identity, as evident by the ability of UNC-30/PITX to control expression of GABA synthesis proteins (*Eastman et al., 1999*; *Jin et al., 1994*) (*Figure 7F*). Similarly, recruitment of UNC-3 enabled 'VD-like' cells to become cholinergic MNs. In this 'new' cholinergic cell type, UNC-3 exerts a dual role: it antagonizes the ability of LIN-39 to activate VD-specific genes, and also synergizes with LIN-39 to co-activate cholinergic MN terminal identity genes (*Figure 9D*). We hope the strategy described here of a terminal selector preventing a Hox transcriptional switch may provide a conceptual framework for future studies on terminal identity and evolution of neuronal cell types.

# Materials and methods

### Key resources table

| Reagent type (species) or resource | Designation | Source or reference | Identifiers | Additional information |
|---|---|---|---|---|
| Gene (*Caenorhabditis elegans*) | *unc-3* | Wormbase | WBGene00006743 | |
| Gene (*Caenorhabditis elegans*) | *unc-30* | Wormbase | WBGene00006766 | |
| Gene (*Caenorhabditis elegans*) | *lin-39* | Wormbase | WBGene00003024 | |
| Gene (*Caenorhabditis elegans*) | *mab-5* | Wormbase | WBGene00003102 | |
| Strain, strain background (*Caenorhabditis elegans*) | *unc-3 (n3435)* | Bob Horvitz (MIT, Cambridge MA) | MT10785 | Null Allele: deletion |
| Strain, strain background (*Caenorhabditis elegans*) | *unc-30 (e191)* | Caenorhabditis Genetics Center | CB845 | Allele: substitution |
| Strain, strain background (*Caenorhabditis elegans*) | *lin-39(n1760)/dpy-17(e164) unc-32(e189) III.* | Caenorhabditis Genetics Center | MT4009 | Null Allele: substitution |
| Strain, strain background (*Caenorhabditis elegans*) | *mab-5 (n1239) III; him-5 (e1490) V* | Caenorhabditis Genetics Center | CB3531 | Allele: substitution |
| Strain, strain background (*Caenorhabditis elegans*) | *him-8 (e1489) IV* | Caenorhabditis Genetics Center | CB1489 | Allele: substitution |

*Continued on next page*

Continued

| Reagent type (species) or resource | Designation | Source or reference | Identifiers | Additional information |
|---|---|---|---|---|
| Strain, strain background (*Caenorhabditis elegans*) | *ieSi57 II; unc-3 (ot837 [unc-3::mNG::AID])* | Caenorhabditis Genetics Center | OH13988 | CRISPR-generated allele |
| Strain, strain background (*Caenorhabditis elegans*) | *lin-39 (kas9 [lin-39::mNG::AID])* | This paper | KRA110 | See Materials and methods, Section Targeted genome editing |
| Strain, strain background (*Caenorhabditis elegans*) | *ieSi57 [eft-3prom::tir1]* | Caenorhabditis Genetics Center | CA1200 | Genotype: *ieSi57 II; unc-119(ed3) III.* |
| Strain, strain background (*Caenorhabditis elegans*) | *ser-2::gfp* | Caenorhabditis Genetics Center | OH2246 | Genotype: *otIs107 I* |
| Strain, strain background (*Caenorhabditis elegans*) | *oig-1::gfp* | Caenorhabditis Genetics Center | OH3955 | Genotype: *pha-1(e2123) III; otEx193* |
| Strain, strain background (*Caenorhabditis elegans*) | *ida-1::gfp* | Caenorhabditis Genetics Center | BL5717 | Genotype: *inIs179 II; him-8(e1489) IV* |
| Strain, strain background (*Caenorhabditis elegans*) | *glr-5::gfp* | Aixa Alfonso (University of Illinois, Chicago IL) | AL270 | Genotype: *icIs270 X* |
| Strain, strain background (*Caenorhabditis elegans*) | *srb-16::gfp* | Caenorhabditis Genetics Center | BC14820 | Genotype:*dpy-5(e907) I; sEx14820* |
| Strain, strain background (*Caenorhabditis elegans*) | *flp-11::gfp* | Caenorhabditis Genetics Center | NY2040 | Genotype: *ynIs40 V* |
| Strain, strain background (*Caenorhabditis elegans*) | *twk-46::gfp* | Caenorhabditis Genetics Center | BC13337 | Genotype: *dpy-5(e907) I; sIs12928 V* |
| Strain, strain background (*Caenorhabditis elegans*) | *ilys-4::tagrfp* | This paper | KRA22 | Genotype: *pha-1(e2123) III; kasEx22* |
| Strain, strain background (*Caenorhabditis elegans*) | *flp-13::gfp* | Caenorhabditis Genetics Center | NY2037 | Genotype: *ynIs37 III* |
| Strain, strain background (*Caenorhabditis elegans*) | *lin-11::mCherry* | Oliver Hobert (Columbia University, New York NY) | OH11954 | Genotype: *lin-11::mCherry + myo-2::GFP V* |
| Strain, strain background (*Caenorhabditis elegans*) | *klp-4::gfp* | Caenorhabditis Genetics Center | BC11799 | Genotype: *dpy-5(e907) I; sEx11799* |
| Strain, strain background (*Caenorhabditis elegans*) | *alr-1::egfp* | Caenorhabditis Genetics Center | OP200 | Genotype: *unc-119(ed3) III; wgIs200 X* |

*Continued*

| Reagent type (species) or resource | Designation | Source or reference | Identifiers | Additional information |
|---|---|---|---|---|
| Strain, strain background (*Caenorhabditis elegans*) | *irx-1::egfp* | Caenorhabditis Genetics Center | OP536 | Genotype: *unc-119(tm4063) III; wgIs536 I* |
| Strain, strain background (*Caenorhabditis elegans*) | *del-1::gfp* | Caenorhabditis Genetics Center | NC138 | |
| Strain, strain background (*Caenorhabditis elegans*) | *acr-2::gfp* | Caenorhabditis Genetics Center | CZ631 | Genotype: *juIs14 IV* |
| Strain, strain background (*Caenorhabditis elegans*) | *unc-129::gfp* | Caenorhabditis Genetics Center | evIs82b | Genotype: *evIs82b IV* |
| Strain, strain background (*Caenorhabditis elegans*) | *dbl-1::gfp* | Caenorhabditis Genetics Center | BW1935 | Genotype: *unc-119(ed3) III; ctIs43 him-5(e1490) V* |
| Strain, strain background (*Caenorhabditis elegans*) | *nca-1::gfp* | Caenorhabditis Genetics Center | BC15028 | Genotype: *dpy-5(e907) I; sEx15028* |
| Strain, strain background (*Caenorhabditis elegans*) | *slo-2::gfp* | Caenorhabditis Genetics Center | BC10749 | Genotype: *dpy-5(e907) I; sEx10749* |
| Strain, strain background (*Caenorhabditis elegans*) | *ttr-39::mCherry* | Caenorhabditis Genetics Center | CZ8332 | Genotype: *juIs223 IV* |
| Strain, strain background (*Caenorhabditis elegans*) | *cho-1::rfp* | Caenorhabditis Genetics Center | OH13646 | Genotype: *pha-1(e2123) III; him-5(e1490) otIs544 V* |
| Strain, strain background (*Caenorhabditis elegans*) | *unc-17::gfp* | Caenorhabditis Genetics Center | LX929 | Genotype: *vsIs48 X* |
| Strain, strain background (*Caenorhabditis elegans*) | *unc-25::gfp* | Caenorhabditis Genetics Center | CZ13799 | Genotype: *juIs76 II* |
| Strain, strain background (*Caenorhabditis elegans*) | *unc-47::mChOpti* | Caenorhabditis Genetics Center | OH13105 | Genotype: *him-5(e1490) otIs564 V* |
| Strain, strain background (*Caenorhabditis elegans*) | *unc-30::gfp* | Caenorhabditis Genetics Center | OP395 | Genotype: *unc-119(tm4063) III; wgIs395* |
| Strain, strain background (*Caenorhabditis elegans*) | *ser-2::rfp* | Mark Alkema (University of Massachusetts, Worcester MA) | AL270 | Genotype: *zfIs8 IV* |
| Strain, strain background (*Caenorhabditis elegans*) | *oig-1(fosmid)::GFP* | Caenorhabditis Genetics Center | OH11809 | Genotype: *otIs450* |

*Continued on next page*

Continued

| Reagent type (species) or resource | Designation | Source or reference | Identifiers | Additional information |
|---|---|---|---|---|
| Strain, strain background (*Caenorhabditis elegans*) | *lin-39::gfp* | Caenorhabditis Genetics Center | OP18 | Genotype: *unc-119(ed3) III; wgIs18* |
| Genetic reagent (*Caenorhabditis elegans*) | *Poig-1_1 kb::gfp* | Oliver Hobert (Columbia University, New York NY) | otEx5993 otEx5994 otEx5995 | |
| Genetic reagent (*Caenorhabditis elegans*) | *Poig-1_1.6 kb::gfp* | This paper | kasEx147 kasEx148 | See Materials and methods |
| Genetic reagent (*Caenorhabditis elegans*) | *Poig-1_2.6 kb_LIN-39 site #3 DEL::gfp* | This paper | kasEx149 kasEx150 | See Materials and methods |
| Genetic reagent (*Caenorhabditis elegans*) | *Poig-1_2.6 kb_LIN-39 site #4 DEL::gfp* | This paper | kasEx151 kasEx152 | See Materials and methods |
| Genetic reagent (*Caenorhabditis elegans*) | *Poig-1_125 bp_::tagrfp* | This paper | kasEx80 kasEx81 kasEx82 | See Materials and methods |
| Genetic reagent (*Caenorhabditis elegans*) | *Poig-1_ LIN-39 site mut 125 bp_::tagrfp* | This paper | kasEx91 kasEx92 kasEx93 | See Materials and methods |
| Genetic reagent (*Caenorhabditis elegans*) | *Punc-3_558bp > lin-39 RNAi + myo-2::gfp* | This paper | kasEx68 kasEx69 kasEx70 kasEx71 kasEx72 | See Materials and methods |
| Genetic reagent (*Caenorhabditis elegans*) | *Punc-3_558bp > unc-3 RNAi + myo-2::gfp* | This paper | kasEx73 kasEx74 kasEx78 kasEx79 | See Materials and methods |
| Genetic reagent (*Caenorhabditis elegans*) | *Punc-3_558bp > lin-39 cDNA OE + myo-2::gfp* | This paper | kasEx35 kasEx36 kasEx37 kasEx76 kasEx77 | See Materials and methods |
| Genetic reagent (*Caenorhabditis elegans*) | *Punc-47 > unc-3 cDNA + myo-2::gfp* | This paper | kasEx75 | See Materials and methods |
| Genetic reagent (*Caenorhabditis elegans*) | *Pcho-1_280bp > lin-39 cDNA OE + myo-2::gfp* | This paper | kasEx38 kasEx39 kasEx41 | See Materials and methods |
| Genetic reagent (*Caenorhabditis elegans*) | *lin-39 fosmid WRM0616aE11 + myo-2::gfp* | This paper | kasEx33 kasEx34 | See Materials and methods |
| Antibody | anti-Myc (Rabbit polyclonal) | Abcam | #ab9106; RRID:AB_307014 | 1:1000 dilution |
| Antibody | anti-Flag (Mouse monoclonal) | Sigma | #F3165; RRID:AB_259529 | 1:1000 dilution |
| Antibody | anti-Flag (Rabbit polyclonal) | Sigma, | #SAB4301135; RRID: AB_2811010 | 1:1000 dilution |
| Antibody | Clean-Blot IP Detection Reagent (Mouse monoclonal) | Thermo Fisher | #21230; RRID: AB_2576514 | See Materials and methods |
| Antibody | Flag antibody coated beads (Mouse monoclonal) | Sigma, | #A2220; RRID:AB_10063035 | See Materials and methods |

*Continued on next page*

*Continued*

| Reagent type (species) or resource | Designation | Source or reference | Identifiers | Additional information |
|---|---|---|---|---|
| Antibody | anti-FLAG M2 magnetic beads (Mouse monoclonal) | Sigma-Aldric | M8823; RRID: AB_2637089 | See Materials and methods |
| Recombinant DNA reagent | pcDNA 3.1(+)-C-Flag (Plasmid) | Genscript | pcDNA 3.1(+) | C-terminus Flag-tagged UNC-3 |
| Recombinant DNA reagent | pcDNA 3.1(+)-N-Myc (Plasmid) | Genscript | pcDNA 3.1(+) | N-terminus Myc-tagged LIN-39 |
| Recombinant DNA reagent | Fosmid clone WRM0616aE11 | Source BioScience | WRM0616aE11 | *lin-39::GFP* fosmid clone |
| Commercial assay or kit | Gibson Assembly Cloning Kit | NEB | #5510S | |
| Commercial assay or kit | QIAquick PCR Purification Kit | QIAGEN | #28104 | |
| Commercial assay or kit | Ampure XP beads | Beckman Coulter Life Sciences | A63881 | |
| Commercial assay or kit | TOPO XL-2 Complete PCR Cloning Kit | Thermo Fisher | K8050 | |
| Chemical compound, drug | Auxin (indole-3-acetic acid) | Alfa Aesar | #10196875 | |
| Software, algorithm | ZEN | ZEISS | Version 2.3.69.1000, Blue edition | RRID:SCR_013672 |
| Software, algorithm | Image J | Image J | Version 1.52i | RRID:SCR_003070 |
| Software, algorithm | RStudio | RStudio | Version 1.2.5001 | |
| Software, algorithm | Adobe Photoshop CS6 | Adobe | Version 13.0 × 64 | |
| Software, algorithm | Adobe Illustrator CS6 | Adobe | Version 16.0.0 × 64 | |

## *C. elegans* strains

Worms were grown at 15 ˚C, 20 ˚C or 25 ˚C on nematode growth media (NGM) plates seeded with bacteria (*E.coli* OP50) as food source (*Brenner, 1974*).

## Forward genetic screen

EMS mutagenesis was performed on *unc-3 (n3435); ynIs40 [flp-11::GFP]* animals using standard procedures (*Kutscher and Shaham, 2014*). Mutagenized L4 animals were visually screened at a dissecting fluorescence microscope for changes in *flp-11::GFP* expression in VNC MNs. One mutant (*kas1*) was retrieved.

## Generation of transgenic reporter animals

Reporter gene fusions for *cis*-regulatory analysis of terminal identity genes were made using either PCR fusion (*Hobert, 2002*) or Gibson Assembly Cloning Kit (NEB #5510S). Targeted DNA fragments were fused (ligated) to *tagrfp* coding sequence, which was followed by *unc-54 3' UTR*. The TOPO XL PCR cloning kit was used to introduce the PCR fusion fragments into the pCR-XL-TOPO vector (Invitrogen). Mutations on LIN-39 motifs were introduced via mutagenesis PCR. The product DNA fragments were either injected into young adult *pha-1(e2123)* hermaphrodites at 50 ng/μl using *pha-1* (pBX plasmid) as co-injection marker (50 ng/μl) and further selected for survival, or injected into young adult N2 hermaphrodites at 50 ng/μl (plus 50 ng/μl pBX plasmid) using *myo-2::gfp* as co-injection marker (3 ng/μl) and further selected for GFP signal.

The fosmid clone WRM0616aE11 (genomic region: III:7519128..7554793) (Source BioScience) that contains the entire *lin-39* locus was linearized by restriction enzyme digestion, mixed with sonicated

bacterial genomic DNA (12 ng/μl) and injected into young adult N2 hermaphrodites at 15 ng/μl using *myo-2::gfp* as co-injection marker (3 ng/μl).

## Generation of transgenic animals for RNAi or over-expression

The cDNA (for over-expression) or the exon-rich genomic region (for RNAi) of *unc-3* and *lin-39* were amplified by PCR and then ligated to cholinergic (*cho-1, unc-3*) or GABAergic (*unc-47*) MN promoters using Gibson Assembly Cloning Kit (NEB #5510S). For *unc-3* RNAi, we targeted exons 2–5 with the following primers: FRW: GTCTGTAAAAGATGAGAACCAGCGG, RVS: CTGTCAATAATAACTGGATCGCTGG. For *lin-39* RNAi, we targeted exons 3–5 with the following primers: FRW: gtggtcaaactccgaacttaaagtg, RVS: gaaggggcgagaaatgtgtgataac. For over-expression constructs, DNA products were purified using a PCR purification protocol (QIAGEN), and then injected into young adult WT hermaphrodites at 50 ng/μl together with 50 ng/μl pBS plasmid (filler DNA) and 3 ng/μl of *myo-2::gfp* (co-injection marker). For RNAi constructs, complementary sense and anti-sense exon-rich genomic regions of *unc-3* and *lin-39* were PCR purified and injected into young adult WT or *unc-3 (n3435)* hermaphrodites each at 100 ng/μl with *myo-2::gfp* as co-injection marker (3 ng/μl) following previously established procedures (*Esposito et al., 2007*).

## Targeted genome engineering

To generated the *lin-39 (kas9 [lin-39::mNG::AID])* allele, CRISPR/Cas9 genome editing was employed to introduce the *mNG::3xFLAG::AID* cassette into the *lin-39* gene locus before the stop codon. Micro-injection, selection and strain establishment were performed as previously described (*Dickinson et al., 2015*).

## Temporally-controlled protein degradation

In the presence of TIR1, AID-tagged proteins are conditionally degraded when exposed to auxin in the presence of TIR1 (*Zhang et al., 2015*). Animals carrying auxin-inducible alleles of *lin-39 (kas9 [lin-39::mNG::AID])* or *unc-3 (ot837 [unc-3::mNG::AID])* (*Kerk et al., 2017*) were crossed with ieSi57 [*eft-3prom::tir1*] animals that express TIR1 ubiquitously. Auxin (indole-3-acetic acid [IAA]) was dissolved in ethanol (EtOH) to prepare 400 mM stock solutions which were stored at 4°C for up to one month. NGM agar plates with fully grown OP50 bacteria were coated with auxin solution to a final concentration of 4 mM, and allowed to dry overnight at room temperature. To induce protein degradation, worms of the experimental strains were transferred onto auxin-coated plates and kept at 20°C. As control, worms were transferred onto EtOH-coated plates instead. Auxin solutions, auxin-coated plates, and experimental plates were shielded from light.

## Microscopy

Worms were anesthetized using 100 mM of sodium azide (NaN$_3$) and mounted on a 4% agarose pad on glass slides. Images were taken using an automated fluorescence microscope (Zeiss, Axio Imager. Z2). Acquisition of several z-stack images (each ~1 μm thick) was taken with Zeiss Axiocam 503 mono using the ZEN software (Version 2.3.69.1000, Blue edition, RRID:SCR_013672). Representative images are shown following max-projection of 1–8 μm Z-stacks using the maximum intensity projection type. Image reconstruction was performed using Image J software (RRID:SCR_003070; *Schindelin et al., 2012*).

## Chromatin immunoprecipitation (ChIP)

ChIP assay was performed as previously described (*Yu et al., 2017*; *Zhong et al., 2010*) with the following modifications. Synchronized *unc-3 (n3435); lin-39 (kas9 [lin-39::mNG::3xFLAG::AID]* worms at L1 stage were cultured on 10 cm plates seeded with OP50 at 20°C overnight. Early L3 worms were cross-linked and resuspended in FA buffer supplemented with protease inhibitors (150 mM NaCl, 10 μl 0.1 M PMSF, 100 μl 10% SDS, 500 μl 20% N-Lavroyl sarsosine sodium, 2 tablets of cOmplete ULTRA Protease Inhibitor Cocktail [Roche Cat.# 05892970001] in 10 ml FA buffer). The sample was then sonicated using a Covaris S220 at the following settings: 200 W Peak Incident Power, 20% Duty Factor, 200 Cycles per Burst for 60 s. Samples were transferred to centrifuge tubes and spun at the highest speed for 15 min. The supernatant was transferred to a new tube, and 5% of the material was saved as input and stored at −20°C. Twenty (20) μl of equilibrated anti-FLAG M2 magnetic

beads (Sigma-Aldrich M8823) were added to the remainder. The *lin-39 (kas9 [lin-39::mNG::3xFLAG::AID])* CRIPSR-generated allele was used in order to precipitate the immunocomplex comprising the endogenous LIN-39 protein and the bound DNA. The immunocomplex was incubated and rotated overnight at 4°C. On the next day, the beads were washed at 4°C twice with 150 mM NaCl FA buffer (5 min each), once with 1M NaCl FA buffer (5 min). The beads were transferred to a new centrifuge tube and washed twice with 500 mM NaCl FA buffer (10 min each), once with TEL buffer (0.25 M LiCl, 1% NP-40, 1% sodium deoxycholate, 1 mM EDTA, 10 mM Tris-HCl, pH 8.0) for 10 min, twice with TE buffer (5 min each). The immunocomplex was then eluted in 200 μl elution buffer (1% SDS in TE with 250 mM NaCl) by incubating at 65°C for 20 min. The saved input samples were thawed and treated with the ChIP samples as follows. One (1) μl of 20 mg/ml proteinase K was added to each sample and the samples were incubated at 55°C for 2 hr and then at 65°C overnight (12–20 hr) to reverse cross-link. The immunoprecipitated DNA was purified with Ampure XP beads (A63881) according to manufacturer's instructions. Library preparation and Illumina sequencing was performed at the Genomics Core facility of the University of Chicago. The LIN-39 ChIP-Seq data on wild-type animals were generated by the modENCODE project (RRID:SCR_006206).

## Real-time quantitative PCR (qPCR) analysis of ChIP DNA

ChIP was performed on *unc-3 (n3435); wgIs18 (lin-39 $^{fosmid}$::GFP)* animals as described above. qPCR analysis of ChIP DNA was performed to probe enrichment of predicted LIN-39 binding sites at four target genes (*acr-2, dbl-1, unc-129, lin-39*). Three biological replicates were included. The primers used are provided in 5′−3′ orientation: *acr-2* LIN-39 site (FRW: acattcgcaccaacaaagcg; RVS: aaaggacggacccaacagac), *acr-2* 3′ UTR (FRW: tttcagcgccacatgtgtttg; RVS: attgcctagtgattctgagtagagg), *dbl-1* LIN-39 site (FRW: gcacaatccctcgggatcaa; RVS: TAAGTTTTGCGCTGCTGCTG), *dbl-1* 3′ UTR (FRW: atacccgcttctatgtcgcc; RVS: ccgtgacacattgcaccaaa), *unc-129* LIN-39 site (FRW: attcgtgtctcgcagggaac; RVS: atagaggaaccggcaaaggtg), *unc-129* 3′ UTR (FRW: ttctgtctgtacatcttccctacc; RVS: tttgccaagaaacaaagagagcag), lin-39 LIN-39 site (FRW: gacgtctccctctttctcctc; RVS: tccgctttctgagactcac), *lin-39* 3′UTR (FRW: gttcaagaaaaatattgtgcgttcc; RVS: cattttcgctcgaactgatgga). The amplification was conducted in a QuantStudio three using the Power SYBR Green PCR Master Mix (ThermoFisher Cat.# 4367659), with the following program: Step 1: 95°C for 10 min; Step 2: 95°C for 15 s; Step 3: 60°C for 1 min. Repeat steps 2–3 for 40 times.

## Motor neuron identification

Motor neuron (MN) subtypes were identified based on combinations of the following factors: (a) co-localization with fluorescent markers with known expression pattern, (b) invariant cell body position along the ventral nerve cord, or relative to other MN subtypes, (c) MN birth order, and (d) number of MNs that belong to each subtype.

## Bioinformatic analysis

To predict the UNC-3 binding site (COE motif) in the *cis*-regulatory region of *unc-129, del-1, acr-2, unc-77* and *slo-2*, we used the MatInspector program from Genomatix (*Cartharius et al., 2005*) (RRID:SCR_008036). The Position Weight Matrix (PWM) for the LIN-39 binding site is catalogued in the CIS-BP (Catalog of Inferred Sequence Binding Preferences database) (*Weirauch et al., 2014*). To identify putative LIN-39 sites on the *cis*-regulatory regions of *unc-129, del-1, acr-2, unc-77, slo-2, oig-1,* and *ser-2,* we used FIMO (Find Individual Motif Occurrences)(*Grant et al., 2011*), which is one of the motif-based sequence analysis tools of the MEME (Multiple Expectation maximization for Motif Elicitation) bioinformatics suite (http://meme-suite.org/). To predict the binding site for the transcription factor UNC-30, we performed FIMO analysis using the UNC-30 binding motif (WNTAATCHH) described in *Cinar et al. (2005)*. The p-value threshold for the analysis was set at p<0.005.

## Automated worm tracking

Worms were maintained as mixed stage populations by chunking on NGM plates with *E. coli* OP50 as the food source. The day before tracking, 30–40 L4 larvae were transferred to a seeded NGM plate and incubated at 20°C for approximately 24 hr. Five adults are picked from the incubated plates to each of the imaging plates (see below) and allowed to habituate for 30 min before

recording for 15 min. Imaging plates are 35 mm plates with 3.5 mL of low-peptone (0.013% Difco Bacto) NGM agar (2% Bio/Agar, BioGene) to limit bacteria growth. Plates are stored at 4°C for at least two days before use. Imaging plates are seeded with 50 µl of a 1:10 dilution of OP50 in M9 the day before tracking and left to dry overnight with the lid on at room temperature.

## Behavioral feature extraction and analysis

All videos were analyzed using Tierpsy Tracker (*Javer et al., 2018a*) to extract each worm's position and posture over time. These postural data were then converted into a set of behavioral features as previously described (*Javer et al., 2018b*). From the total set of features, we only considered 48 that are related to midbody posture and motion, as well as the midbody width (see *Supplementary file 3* for feature descriptions and their average values for each strain). For each strain comparison, we performed unpaired two-sample t-tests independently for each feature. The false discovery rate was controlled at 5% across all strain and feature comparisons using the Benjamini Yekutieli procedure (*Kim and van de Wiel, 2008*). The p-value threshold to control the false discovery rate at 0.05 is 0.0032.

## Cloning, western blot, and immunoprecipitation

UNC-3, and LIN-39 cDNAs were cloned into the mammalian expression vectors pcDNA 3.1(+)-C-Flag plasmid and the pcDNA 3.1(+)-N-Myc plasmid by GeneScript, to generate C-terminus Flag-tagged UNC-3 and N-terminus Myc-tagged LIN-39. The constructs were verified by sequencing at the sequencing core facility of University of Chicago. The tagged proteins were expressed in HEK293 cells. Protein expression was detected by standard western blot. Expression of Myc tagged LIN-39 was detected using anti-Myc (Abcam, #ab9106, RRID:AB_307014), expression of Flag-tagged UNC-3 in the total cell lysate was detected using mouse anti-Flag (Sigma, #F3165, RRID:AB_259529), expression of Flag-tagged UNC-3 in the IP was detected using rabbit anti-Flag (Sigma, #SAB4301135, RRID:AB_2811010). Immunoprecipitation of Flag-tagged UNC-3 was performed using Flag antibody coated beads (Sigma, #A2220). For the IP, the 'Clean-Blot IP Detection Reagent' (Thermo Fisher, #21230) was used as secondary antibody.

## Quantification of fluorescence intensity

Images of worms carrying the *lin-39::mNG::3xFLAG::AID* or *unc-3::mNG:: 3xFLAG::AID* alleles were taken on the same slide with the same camera settings at the same development stage. Acquisition of four z-stack images (each 0.53 µm thick) covering the middle portion of targeted MN cell bodies was taken with Zeiss Axiocam 503 mono using the ZEN software (Version 2.3.69.1000, Blue edition, RRID:SCR_013672). Image reconstruction was performed using Image J software following average-projection the Z-stacks using the average intensity projection protocol. The chosen cells for quantification of mNG fluorescence intensity for both genotypes are the same 10 cholinergic MNs: AS2, DB3, DA2, VA3, VB4, AS3, DA3, VA4, VB5 and DB4. Targeted cell areas were manually selected with minimum background as region of interest (ROI) and the total fluorescence Intensity was measured, calculated, and then represented by Image J as Integrated Density – IntDen (ROI). Background was additionally selected and IntDen (Background) was calculated. The net fluorescence intensity increase is represented as NeIncr = IntDen(ROI)/Area(ROI) – IntDen (Background)/Area (Background). 12 NetIncrs of both genotypes were calculated and data were normalized by dividing the Median of NetIncr (*lin-39::mNG::3xFLAG::AID*) for better contrast and presented as arbitrary units (a.u).

## Statistical analysis

For data quantification, graphs show values expressed as mean ± standard deviation (STDV). The statistical analyses were performed using the unpaired t-test (two-tailed). Calculations were performed using the Evan's Awesome A/B Tools online software (https://www.evanmiller.org/ab-testing/t-test.html). Differences with p<0.05 were considered significant. Quantifications are provided in the form of box-and-whisker plots (Tukey boxplot) with individual data point dot-plotted. In all boxplots, middle horizontal line represents the median value (equals to Q2). The box illustrates the interquartile range (IQR), that is from Q1 to Q3. The upper limit indicates either the maximum

value if maximum <Q3 + 1.5*IQR, or the value that is not higher than Q3 + 1.5*IQR. Similarly, the lower limit indicates either the minimum value or the value that is not lower than Q1 - 1.5*IQR.

## Acknowledgements

We thank the Caenorhabditis Genetics Center (CGC), which is funded by NIH Office of Research Infrastructure Programs (P40 OD010440), for providing strains. We thank Anthony Osuma, Melanie Le Gouez, and Minhkhoi Nguyen for generating *lin-39* and *oig-1* reporter strains. We are grateful to Oliver Hobert, Elizabeth Heckscher, Robert Carillo, Catarina Catela, and Daniele Canzio for comments on this manuscript. This work was funded by an NINDS grant (R00NS084988) and a Whitehall Foundation grant to PK.

## Additional information

### Funding

| Funder | Grant reference number | Author |
|---|---|---|
| National Institute of Neurological Disorders and Stroke | K99/R00: Pathway to Independence Award | Paschalis Kratsios |
| Whitehall Foundation | 2017-12-50 | Paschalis Kratsios |

The funders had no role in study design, data collection and interpretation, or the decision to submit the work for publication.

### Author contributions

Weidong Feng, Conceptualization, Data curation, Formal analysis, Investigation, Methodology, Writing—review and editing; Yinan Li, Pauline Dao, Formal analysis, Validation, Investigation, Visualization, Writing—review and editing; Jihad Aburas, Priota Islam, Benayahu Elbaz, Anna Kolarzyk, Investigation; André EX Brown, Project administration, Writing—review and editing; Paschalis Kratsios, Conceptualization, Supervision, Funding acquisition, Investigation, Writing—original draft, Project administration, Writing—review and editing

### Author ORCIDs

Paschalis Kratsios (iD) https://orcid.org/0000-0002-1363-9271

### Decision letter and Author response

Decision letter https://doi.org/10.7554/eLife.50065.sa1
Author response https://doi.org/10.7554/eLife.50065.sa2

## Additional files

### Supplementary files

• Supplementary file 1. UNC-3 binding sites (COE motifs) are not found in the *cis*-regulatory region of VD- and VC-expressed terminal identity genes.

• Supplementary file 2. LIN-39/Hox targets in cholinergic and GABAergic (VD) motor neurons. Asterisk (*) highlights novel LIN-39 targets; N. D: Not Determined. The selected *cis*-regulatory regions are LIN-39 ChIP-seq peaks that fall within the DNA sequence used for our reporter gene constructs (except for *del-1*). The UNC-3 binding sites (COE motifs 23 bp) have been previously described in *Kratsios et al. (2012)*. The LIN-39 binding sites were predicted by a FIMO search (p<0.005). The UNC-30 binding site on *ser-2* locus was predicted by a FIMO search. The UNC-30 site on *oig-1* was experimentally validated in *Howell et al. (2015)*.

• Supplementary file 3. Locomotion features assessed by automated worm tracking analysis.

• Transparent reporting form

Data availability
All data generated or analysed during this study are included in the manuscript and supporting files.

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
