## [Decision Letter]

**Acceptance summary:**

Your paper establishes the molecular mechanism used to determine and maintain terminal identity of cholinergic motor neurons in *C. elegans*. Your demonstration of the dual function of UNC-3 (Collier/Ebf) in directly promoting identity features of cholinergic motor neurons and in preventing terminal features of other ventral cord neuron types is striking. The role of the Hox protein LIN-39 that interacts with UNC-3 explains this dual role. Your data therefore shows that UNC-3 and LIN-39 co-activate terminal identity features of cholinergic motor neurons.

**Decision letter after peer review:**

Thank you for submitting your article "A terminal selector prevents a Hox transcriptional switch to safeguard motor neuron identity throughout life" for consideration by *eLife*. Your article has been reviewed by three peer reviewers, and the evaluation has been overseen by a Reviewing Editor and Marianne Bronner as the Senior Editor. The reviewers have opted to remain anonymous.

The reviewers have discussed the reviews with one another and the Reviewing Editor has drafted this decision to help you prepare a revised submission.

The general opinion of the reviewers is that the paper represents an important contribution to the field and presents concepts that should be of general interest to those studying neural specification in general and production of neurotransmitters in particular. However, while the regulatory network is clearly presented in the paper, there are some significant questions that remain and need to be addressed before the paper can be considered for publication.

In particular, a technical problem appears to be in the ChIP-PCR that you performed. It seems that are available ChIP-seq data that would be more valuable if they are of quality. If not, you should sequence your ChIP with Lin-39 and redo them in a *unc-3* mutant. This should also complemented by a detailed description of the *unc-3* phenotype, as suggested by reviewer #2.

The other significant issue resides with the biochemistry in order to elucidate how these proteins interact with the promoters to regulate cell fate and neurotransmitters. The interaction between Lin39 and Unc-3 should be clarified.

Furthermore, the paper would gain by being presented in a simpler way so it would reach a wider audience. We also ask that you remove unsupported claims, as described by the reviewers below.

Reviewer #1:

The manuscript by Feng et al. examines mechanisms through which UNC-3 transcription factor induces expression of genes expressed in cholinergic motor neurons while at the same time it suppresses expression of VD GABAergic motor neuron markers. The study provides a detailed genetic dissection of this problem, concluding that derepression of VD identity in the absence of UNC-3 is dependent on LIN-39 transcription factor. Interestingly, UNC-3 does not repress LIN-39 transcriptionally, instead it diverts its activity from VD target genes to cholinergic VNC motor neuron identity genes. The authors then investigate whether LIN-39 is sequestered away from VD enhancers by UNC-3 to regulatory elements associated with cholinergic motor neuron genes, as has been suggested for many other transcription factors (e.g. Mef2D-Crx interactions in retina (Andzelm et al., 2015) or Isl1-Lhx3/Onecut1 interaction in mammalian motor neurons (Rhee et al., 2016)). However, biochemical tests do not support this model. The authors do not detect physical interaction between UNC-3 and LIN-39, and they suggest that LIN-39 in not a limiting factor as it is not displaced from its binding sites in the absence of UNC-3. At the same time, they contradict this conclusion by demonstrating that VD genes can be induced in the context of cholinergic motor neurons by simply increasing the dosage of LIN-39.

While the genetic analysis and characterization of mutant phenotypes is rigorous and convincing, I am less convinced by the chromatin immunoprecipitation data. Mainly, their ChIP-PCR analysis lacks proper negative controls (regions not bound by transcription factors). More importantly, ChIP-PCR is an inferior method for the analysis of transcription factor binding, prone to false positives and general misinterpretation of transcription factor binding data. Surprisingly, the authors take minimal advantage of the existing LIN-39 ChIP-seq data. If they believe the data are of good quality, they should rely on them to demonstrate binding to the discovered enhancers. If they believe that their data are better, they should sequence their material and include it in the manuscript. In addition, they should perform ChIP-seq analysis in *unc-3* animals to substantiate their claim that LIN-39 does not relocate from cholinergic enhancers to VD enhancers.

Overall, the manuscript is long and dense. The authors should make an attempt to streamline it and simplify it by focusing on the most important points.

In several places the authors state that a "uniform" population of cholinergic motor neurons gives rise to distinct VC and VD neuronal identities in *unc-3* animals. However, VNC cholinergic neurons are not a uniform population as they are composed of five different subtypes. Furthermore, the data clearly show that VC neurons are mainly derived from AS neurons while VD neurons are primarily derived from DA/B and VA/B neurons.

It appears that all tested sites listed in Supplementary file 2 show LIN-39 binding – this raises the concern that the ChIP is globally enriched for DNA and any tested regions will be positive. It is essential to include negative genomic regions in ChIP-PCR experiments and in general it would be preferable to perform ChIP-seq analysis. The authors should show existing LIN-39 ChIP-seq tracks for each of the examined regulatory regions.

The authors conclude that LIN-39 is not a limiting factor. How do they interpret the LIN-39 overexpression results that leads to the activation of VD markers in VNC cholinergic neurons?

The authors should provide better description of LIN-39 expression pattern and describe its mutant phenotype in detail.

The authors present their model as revealing completely new mechanisms of regulation of gene expression. However, cell type-specific transcription factor binding to distinct enhancers is a well-known phenomenon with several publications detailing how interactions with different transcription factors result in relocation of TF complexes from one region to another. The authors should present their work in the context of this literature.

Reviewer #2:

This work from the Kratsios lab aims to understand how neurons acquire and importantly maintain their fate. It takes advantage of the well-studied and characterized *C. elegans* nervous system, focusing on motor neurons. The wealth of genetic, cellular and molecular tools makes *C. elegans* an appropriate model system. Additionally, motor neurons are particularly interesting cell type because they acquire and maintain a generic motor neuron fate, plus an appropriate motor neuron subtype. Thus, the topic is relevant and addressed with appropriate tools.

Previous work, with a significant contribution by Kratsios, established that the motor neuron terminal selector *unc-3* directly activates and maintains a sub-set of motor neuron fates. These findings are confirmed and expanded in this manuscript. Of note, the degron experiment presented here is a nice confirmation of this hypothesis. Similar to other species, Hox genes coordinate their activity to establish distinct motor neurons fates. In this work, *lin-39* recovered though a screen and *mab-5* as candidate Hox genes regulating terminal genes. Again, there is a nice confirmation of this *lin-39* as terminal selector hypothesis by ChIP and the Lin-39 degron allele.

One interesting take home from this work is the underlying gene regulatory network. Not all motor neurons acquire the same "default state" upon *unc-3* removal and divide themselves into 3 groups based on the set of analyzed makers: VD, VC or undetermined. These results suggest a heterogeneous underlying regulatory network controlling motor neuron fate that might be canalized by *unc-3* activity. Because it is at the crux of the phenotype, the author should be very clear describing the phenotype. Is this an expressivity issue or it is 100% penetrant with high expressivity in always the same cells across animals? or is not fully penetrant or with variable expressivity across motor neurons? Depending on the phenotype, combining all motor neurons in a single group might not be the appropriate data representation. Additionally, it will significantly change the conclusions that can be derived from the data.

At large, this work revolves around the combinatorial control of cell fate. It further dissects the *unc-3*/Hox dependent regulation of motor neuron fate. Activation of neuronal type-specific features and preventing the expression of different cell fates is a well-described transcription factor activity during chick and mouse motor neuron development. I am confused by the strong emphasis on "transcriptional switch". It is stressed as a novel insight but there is not tangible mechanistic evidence for it. Moreover, the phenotypes are changes in gene expression with repression and activation of terminal genes, not a complete switch of fates as far as I can tell. At the mechanistic level of gene regulation by transcription factors, there is no evidence of *unc-3* or *lin-39* switching binding targets or DNA motif preference. Additionally, transcription factor target switch by changing binding partners has been extensively described, even during motor neuron development.

Finally, there is a trend to make claims that are not strictly supported by the data or a lack of precision describing phenotypes. In my opinion, this work has a few confirmatory results, the significant claim is not supported by the data and thus it does not provide a significant conceptual nor technical leap for *eLife*.

Reviewer #3:

This paper investigated the molecular mechanism used to determine and maintain terminal identity features of cholinergic motor neurons in the *C. elegans* ventral nerve cord. The authors showed dual function of the terminal selector UNC-3 (Collier/Ebf) in directly promoting identity features of cholinergic motor neurons and indirectly preventing expression of terminal features of three other ventral cord neuron types (VD, VC, CA). Through an unbiased genetic screen, the authors identified the Hox protein LIN-39 (Scr/Dfd/Hox4-5) to be required for ectopic expression of VD and VC features in *unc-3* mutants. In summary, the data presented in the paper supports the dual role of UNC-3: UNC-3 and LIN-39 directly co-activate terminal identity features of cholinergic motor neurons; UNC-3 antagonizes the LIN-39's ability of activating terminal features of alternative neuronal identities.

The paper is well written. However, some points should be addressed or clarified.

- It has been previously shown that *unc-3* and *lin-39* in act synergistically to activate expression of cholinergic motor neuron terminal identity genes (Kratsios et al., 2017). This study identified a new role of *unc-3* in antagonizing the LIN-39's function of activating terminal features of alternative neuronal identities. The authors should perform additional experiments to provide a mechanistic insight into how UNC-3 prevents LIN-39-mediated transcriptional switch for inducing expression of alternative identity features.

- The authors showed data to indicate that *lin-39* and *unc-30* act in parallel to co-activate VD terminal identity genes. However, the authors also showed that ectopic expression of VD markers is not dependent on UNC-30. The contradictory results should be discussed. Does the data suggest that *lin-39* acts with yet-to-be identified factor(s) to activate ectopic expression of VD features in *unc-3* mutants?

---

## [Author Response]

The general opinion of the reviewers is that the paper represents an important contribution to the field and presents concepts that should be of general interest to those studying neural specification in general and production of neurotransmitters in particular. However, while the regulatory network is clearly presented in the paper, there are some significant questions that remain and need to be addressed before the paper can be considered for publication.In particular, a technical problem appears to be in the ChIP-PCR that you performed. It seems that are available ChIP-seq data that would be more valuable if they are of quality. If not, you should sequence your ChIP with Lin-39 and redo them in a unc-3 mutant. This should also complemented by a detailed description of the unc-3 phenotype, as suggested by reviewer #2.The other significant issue resides with the biochemistry in order to elucidate how these proteins interact with the promoters to regulate cell fate and neurotransmitters. The interaction between Lin39 and Unc-3 should be clarified.Furthermore, the paper would gain by being presented in a simpler way so it would reach a wider audience. We also ask that you remove unsupported claims, as described by the reviewers below.

We thank the reviewers for their constructive comments. The revised manuscript addresses all of the comments and, as suggested, contains: (a) available LIN-39 ChIP-seq data, (b) new data we generated for LIN-39 ChIP-seq in *unc-3* mutants, (c) new ChIP-PCR data with appropriate negative controls, (d) a detailed description of *unc-3* and *lin-39* mutant phenotypes and expression patterns, (e) new data that help clarify how LIN-39 interacts with *cis*-regulatory elements to drive expression of terminal identity genes in wild-type cholinergic MNs and VD-specific genes in *unc-3*-depleted MNs. Lastly, we have corrected two claims (cholinergic MNs are not a uniform population, LIN-39 is a rate-limiting factor) and streamlined the text throughout, so that the paper would reach a wider audience.

Reviewer #1:The manuscript by Feng et al. examines mechanisms through which UNC-3 transcription factor induces expression of genes expressed in cholinergic motor neurons while at the same time it suppresses expression of VD GABAergic motor neuron markers. The study provides a detailed genetic dissection of this problem, concluding that derepression of VD identity in the absence of UNC-3 is dependent on LIN-39 transcription factor. Interestingly, UNC-3 does not repress LIN-39 transcriptionally, instead it diverts its activity from VD target genes to cholinergic VNC motor neuron identity genes. The authors then investigate whether LIN-39 is sequestered away from VD enhancers by UNC-3 to regulatory elements associated with cholinergic motor neuron genes, as has been suggested for many other transcription factors (e.g. Mef2D-Crx interactions in retina (Andzelm et al., 2015) or Isl1-Lhx3/Onecut1 interaction in mammalian motor neurons (Rhee et al., 2016)). However, biochemical tests do not support this model. The authors do not detect physical interaction between UNC-3 and LIN-39, and they suggest that LIN-39 in not a limiting factor as it is not displaced from its binding sites in the absence of UNC-3. At the same time, they contradict this conclusion by demonstrating that VD genes can be induced in the context of cholinergic motor neurons by simply increasing the dosage of LIN-39.While the genetic analysis and characterization of mutant phenotypes is rigorous and convincing, I am less convinced by the chromatin immunoprecipitation data. Mainly, their ChIP-PCR analysis lacks proper negative controls (regions not bound by transcription factors). More importantly, ChIP-PCR is an inferior method for the analysis of transcription factor binding, prone to false positives and general misinterpretation of transcription factor binding data. Surprisingly, the authors take minimal advantage of the existing LIN-39 ChIP-seq data. If they believe the data are of good quality, they should rely on them to demonstrate binding to the discovered enhancers. If they believe that their data are better, they should sequence their material and include it in the manuscript. In addition, they should perform ChIP-seq analysis in unc-3 animals to substantiate their claim that LIN-39 does not relocate from cholinergic enhancers to VD enhancers.

We completely agree with the reviewer. Our new set of experiments helped paint a clearer picture of the molecular mechanism underlying the dual role of UNC-3 and support the conclusion that LIN-39 is indeed the rate-limiting factor.

By using available LIN-39 ChIP-seq data, it is evident that LIN-39 directly binds to *cis*-regulatory elements of cholinergic MN identity genes (new Figure 5C, Figure 5—figure supplement 1), corroborating our genetic analysis. By performing ChIP-seq for LIN-39 in *unc-3* mutants, we found that LIN-39 binding is dramatically decreased to these gene loci (new Figure 5C, Figure 5—figure supplement 1). Our new RT-PCR data, in *unc-3* mutants, on these cholinergic MN terminal identity genes further corroborated the ChIP-seq data (new Figure 5—figure supplement 1). Together, these findings suggest that LIN-39 binding on cholinergic MN identity genes depends on UNC-3, raising the possibility that UNC-3 recruits LIN-39 (or diverts LIN-39 activity away) from VD gene promoters. Supporting this scenario, ectopic expression of UNC-3 in VD neurons results in decreased expression of *lin-39*-dependent VD-specific markers (Figure 9C, Figure 9—figure supplement 2C). We thank the reviewer for the suggestion to rely on ChIP-seq data and include additional negative controls (3’UTR) in our RT-PCR analysis, enabling us to clarify the molecular mechanism.

In addition, we quantified the endogenous expression levels of *lin-39* and found it to be expressed at relatively low levels in cholinergic MNs compared to *unc-3* levels, further supporting the conclusion that LIN-39 is the rate-limiting factor. This conclusion is further supported by our gene dosage experiments for *unc-3* and *lin-39* (Figure 9B-C). For example, over-expression of LIN-39 in cholinergic MNs results in activation of VD-specific genes (Figure 9C). Together, our results suggest that the collaboration of UNC-3 with LIN-39 on cholinergic MN identity gene promoters exhausts the limited amount of LIN-39 in the cell and VD genes remain off. However, LIN-39 is released from cholinergic MN gene promoters in the absence of UNC-3 and activates VD genes.

Lastly, we now provide evidence that LIN-39 acts through distinct *cis*-regulatory elements to activate the same VD-specific gene (*oig-1*) in wild-type VD neurons versus VD-like neurons of *unc-3* mutants (new Figure 8, please also see our sixth response to reviewer #2). We have revised the text to include these new findings (Results, subsection “LIN-39 acts through distinct cis-regulatory elements to control *oig-1* expression in VD and VD-like motor neurons”) and also discuss our findings in the context of the Mef2D-Crx and Isl1-Lhx3/Onecut1 studies mentioned by the reviewer (Introduction; Discussion).

Overall, the manuscript is long and dense. The authors should make an attempt to streamline it and simplify it by focusing on the most important points.

We carefully went through the manuscript. The revised text now focuses on the most important points raised in Introduction: (a) post-embryonic requirement of UNC-3 and LIN-39, and (b) dissection of the UNC-3/LIN-39 molecular mechanism of action. In an attempt to streamline and simplify, we carefully edited the text throughout and removed several redundant sentences, as well as an entire paragraph in Discussion.

In several places the authors state that a "uniform" population of cholinergic motor neurons gives rise to distinct VC and VD neuronal identities in unc-3 animals. However, VNC cholinergic neurons are not a uniform population as they are composed of five different subtypes. Furthermore, the data clearly show that VC neurons are mainly derived from AS neurons while VD neurons are primarily derived from DA/B and VA/B neurons.

We thank the reviewer for this comment. We have corrected this statement in Introduction and Discussion.

It appears that all tested sites listed in Supplementary file 2 show LIN-39 binding – this raises the concern that the ChIP is globally enriched for DNA and any tested regions will be positive. It is essential to include negative genomic regions in ChIP-PCR experiments and in general it would be preferable to perform ChIP-seq analysis. The authors should show existing LIN-39 ChIP-seq tracks for each of the examined regulatory regions.

Prompted by this comment, we obtained and analyzed the LIN-39 ChIP-seq data (from modENCODE), which were performed on wild-type animals at L3, a stage at which the defects we discovered on cholinergic MNs of *lin-39* mutants are present. In the revised manuscript, we include LIN-39 ChIP-seq tracks for all LIN-39 targets (*unc-129, del-1, acr-2, unc-77, slo-2, oig-1, ser-2, flp-11*) revealed from our genetic analysis and shown in Supplementary file 2. This data (Figure 5C, Figure 8, Figure 5—figure supplement 1) show clear LIN-39 binding peaks in the *cis*-regulatory region of all these genes, and no peaks at regions that are normally not bound by transcription factors (e.g., 3’ UTR). Moreover, the LIN-39 ChIP-seq peaks align with the LIN-39 sites that we identified bioinformatically based on the LIN-39 consensus motif (Supplementary file 2).

Importantly, we functionally tested the requirement of one LIN-39 binding site in the promoter of *oig-1*, a gene normally activated by LIN-39 in VD neurons. Mutation of this site in the context of transgenic *oig-1* reporter animals carrying a minimal *cis*-regulatory fragment (*oig-1_125bp ^LIN-39 site MUT::RFP^*) led to a significant decrease in *rfp* expression in VD neurons (Figure 8A-B). These findings indicate that our functional testing of LIN-39 sites is in agreement with the binding pattern of LIN-39 on the genome based on published ChIP-seq data.

The authors conclude that LIN-39 is not a limiting factor. How do they interpret the LIN-39 overexpression results that leads to the activation of VD markers in VNC cholinergic neurons?

Our new ChIP-seq data helped clarify the molecular mechanism, as discussed above. The revised manuscript now states that LIN-39 is a rate-limiting factor (Results, subsection “The LIN-39-mediated transcriptional switch depends on UNC-3 and LIN-39 levels”; Discussion, subsection “UNC-3 determines the function of the rate-limiting factor LIN-39/Hox in cholinergic motor neurons”).

The authors should provide better description of LIN-39 expression pattern and describe its mutant phenotype in detail.

We carried out a detailed analysis to describe with single-cell resolution the expression pattern for the mid-body Hox gene *lin-39* in MNs. Using two reporters (a CRISPR-based and a fosmid [~30kb genomic clone]-based reporter) that faithfully recapitulate the endogenous *lin-39* expression pattern, we found that LIN-39 is co-expressed with UNC-3 in 28 cholinergic MNs (from AS2 at the anterior end of the VNC to DB7 at the posterior end). Importantly, this number is in close agreement with the total number of VD-like (12.1 ± 2.6 [mean ± STDV]) and VC-like (10.5 ± 3.7 [mean ± STDV]) cells observed in *unc-3* mutants. A summary of *lin-39* expression in cholinergic MNs is shown in Figure 4E and Figure 4—figure supplement 2.

Following a similar analysis, we determined the expression of *lin-39* in GABAergic neurons of the nerve cord (seven VD [VD3-VD9] and four DD [DD2-DD5] express *lin-39)*. These data are presented in Figure 7A and Figure 4—figure supplement 2. Lastly, we found *lin-39* to be expressed in all six sex-specific VC neurons. Together, this analysis shows that *lin-39* is expressed in every motor neuron located between AS2 and VD9 (Figure 4E, 7A), consistent with the typical, region-specific expression pattern of Hox genes across species.

The function of *lin-39* in cholinergic and GABAergic MNs was largely unknown. Our manuscript describes multiple terminal identity genes under the direct control of LIN-39 in cholinergic and VD motor neurons. Since these genes code for proteins critical for neuronal function (e.g., ion channels, NT receptor, neuropeptides), it is conceivable that loss of expression of these genes leads to the observed locomotion defects of *lin-39* mutants (Discussion). In the revised manuscript, we describe in detail the *lin-39* mutant phenotype in Results, and made new schematics to summarize the *lin-39* effects on terminal identity in Figures 5D and 7F. In addition, we mention in Results that all MNs that control *C. elegans* locomotion (cholinergic and GABA) are normally generated in *lin-39* mutants. However, the six hermaphrodite-specific neurons (VC) that control egg laying do not survive in *lin-39* null mutants, precluding us from testing in these animals whether LIN-39 is acting at later stages of their development. We bypassed the VC survival issue by using our auxin-inducible *lin-39* allele, and found that *lin-39* is required to maintain the expression of VC-specific terminal identity genes, similar to our findings in cholinergic and GABAergic neurons (Results).

The authors present their model as revealing completely new mechanisms of regulation of gene expression. However, cell type-specific transcription factor binding to distinct enhancers is a well-known phenomenon with several publications detailing how interactions with different transcription factors result in relocation of TF complexes from one region to another. The authors should present their work in the context of this literature.

In the revised manuscript (Introduction; Discussion), we now present our findings in the context of previous literature mentioned by reviewer #1 (Andzelm et al., 2015; Rhee et al., 2016) and reviewer #2 (papers on LIM homeodomain TFs Islet1 and Chx10 in of motor neurons and V2 interneurons, respectively). We thank both reviewers for pointing out these important papers. All these studies focus on the combinatorial logic through which TFs control neuronal cell fate. However, there are several noteworthy differences between these papers and our study, as detailed below.

First, the post-embryonic requirement for the TFs mentioned in these previous studies was not tested. Hence, it remained unclear whether the proposed mechanisms operate transiently during development, or continuously throughout life. In fact, Rhee et al. proposed that, in nascent (very young) mammalian MNs in vitro, a dynamic relay of transient enhancers bound by stage-specific TFs controls cell identity. By temporally inactivating UNC-3 and LIN-39 at post-embryonic stages (Figures 3, 6, 7C) in mature MNs, we provide evidence that both of these proteins are continuously required to safeguard motor neuron terminal identity. Importantly, temporal inactivation of the Hox protein LIN-39 was performed in both WT and *unc-3* mutant backgrounds, indicating that LIN-39 is continuously required to: (i) activate cholinergic MN terminal identity genes in the presence of UNC-3, and (ii) activate VD genes in the absence of UNC-3. Lastly, these experiments advanced our understanding of Hox protein function in the nervous system by uncovering that LIN-39/Hox is required to maintain MN identity (Discussion).

Second, certain aspects of our findings are similar to the seminal study by Andzelm et al. in mouse photoreceptors, indicating an evolutionary conservation of the molecular principle we discovered in *C. elegans*. Andzelm et al. found that CRX and MEF2D co-operate to activate expression of retina-specific enhancers in photoreceptor cells. CRX recruits MEF2D to these retina-specific enhancers, thereby preventing MEF2D to bind to other enhancers. However, this study did not address whether and how these “other enhancers” are controlled by MEF2D in other cell types. We provide a comprehensive analysis of LIN-39 not only in the context of cholinergic MNs where it cooperates with UNC-3, but also in the context of VD neurons where it cooperates with another terminal selector (UNC-30/Pitx). We refer to the Andzelm paper in the text (Introduction; Discussion).

Third, seminal work in the context of mouse and chick motor neurons has shown that different TF interactions result in relocation of TF binding from one region to the other. For example, Isl1, Lhx3 and NLI (also known as MN hexamer complex) bind to hemaxer response elements (HxRE) found in the *cis*-regulatory region of MN genes. In V2a interneurons, Lhx3 is not expressed and Isl1 and NLI form a tetrameric complex (V2 tetramer) that binds to tetramer response element (TeRE) upstream of V2a genes. However, the mechanism through which the MN hexamer and the V2 tetramer prevent expression of alternative identities relies on transcriptional repressors (Hb9, Chx10) downstream of these complexes. Although we cannot exclude the possibility that other, yet-to-be identified repressors may be activated by UNC-3 to prevent expression of alternative identities, our current data suggest a simple mechanism where LIN-39 is the rate-limiting factor present in limited amounts in cholinergic MNs. We present our findings in the context of the MN hexamer and V2a tetramer in Discussion (section entitled “*UNC-3 determines the function of the rate-limiting factor LIN-39/Hox in cholinergic motor neurons*”).

Reviewer #2:This work from the Kratsios lab aims to understand how neurons acquire and importantly maintain their fate. It takes advantage of the well-studied and characterized C. elegans nervous system, focusing on motor neurons. The wealth of genetic, cellular and molecular tools makes C. elegans an appropriate model system. Additionally, motor neurons are particularly interesting cell type because they acquire and maintain a generic motor neuron fate, plus an appropriate motor neuron subtype. Thus, the topic is relevant and addressed with appropriate tools.Previous work, with a significant contribution by Kratsios, established that the motor neuron terminal selector unc-3 directly activates and maintains a sub-set of motor neuron fates. These findings are confirmed and expanded in this manuscript. Of note, the degron experiment presented here is a nice confirmation of this hypothesis.

We thank the reviewer for this comment, which gave us the opportunity to improve the manuscript and explain why our findings significantly expand previous work. As stated in Introduction, our article aims to address two important questions in the field: (a) How do neuron type-specific TFs control neuronal terminal identity? (b) Are these factors continuously required (from development to adult)?

Although previous work showed that UNC-3 is continuously required to activate terminal identity genes in *C. elegans* motor neurons, it was not known whether UNC-3 is continuously required to prevent expression of alternative identity genes. Given precedents in the literature describing that TFs can exert stage-specific functions and act transiently (e.g., Rhee et al., 2016), or even switch targets (e.g., Wyler et al., 2016), it was not a given that UNC-3 is continuously required to prevent expression of terminal identity genes for alternative neuronal identities (VD, VC, CA). Our auxin-inducible protein depletion experiments found this to be the case, providing strong evidence for a dual role of UNC-3 that takes place during development and post-embryonic life. Lastly, we provide key molecular insights into how UNC-3 prevents expression of alternative identity genes, i.e., by antagonizing the ability of the mid-body Hox protein LIN-39 to activate expression of alternative identity genes in cholinergic motor neurons. We have modified the text to describe better these findings (Results, Discussion).

Similar to other species, Hox genes coordinate their activity to establish distinct motor neurons fates. In this work, lin-39 recovered though a screen and mab-5 as candidate Hox genes regulating terminal genes. Again, there is a nice confirmation of this lin-39 as terminal selector hypothesis by ChIP and the Lin-39 degron allele.

Indeed, seminal work in multiple model systems has revealed that Hox proteins control motor neuron fate. However, previous studies largely focused in the early steps of motor neuron development (e.g., progenitor specification, initial specification of motor neuron subtypes, establishment of motor neuron connectivity). In most model systems, the expression pattern and function of Hox proteins during the last steps of motor neuron development and post-embryonic life are largely unknown. In the revised manuscript, we discuss this knowledge gap and present our findings in the context of current Hox literature (Discussion, subsection: “Maintenance of terminal identity features: A new function of Hox proteins in the nervous system”).

Prior to our work, it was unclear whether and how Hox proteins control expression of multiple terminal identity genes (e.g., ion channels, NT receptors, neuropeptides) in motor neurons, and, perhaps most important, whether Hox proteins are required at post-embryonic stages to maintain expression of terminal identity genes in motor neurons. We note that these 2 criteria (broad and direct regulation of terminal identity genes by a TF and continuous requirement for a TF) must be fulfilled for a gene to be called a terminal selector, but previous studies did not address these criteria.

Our study identified multiple terminal identity genes as LIN-39 (Hox) targets in different motor neurons (six targets in cholinergic MNs, five targets in VD neurons, one target in VC neurons). Moreover, ChIP-seq for LIN-39 indicates direct control for these genes. Lastly, through embryonic and post-embryonic removal of LIN-39, we show that this Hox protein is required to induce during development and maintain throughout post-embryonic life motor neuron terminal identity.

One interesting take home from this work is the underlying gene regulatory network. Not all motor neurons acquire the same "default state" upon unc-3 removal and divide themselves into 3 groups based on the set of analyzed makers: VD, VC or undetermined. These results suggest a heterogeneous underlying regulatory network controlling motor neuron fate that might be canalized by unc-3 activity. Because it is at the crux of the phenotype, the author should be very clear describing the phenotype. Is this an expressivity issue or it is 100% penetrant with high expressivity in always the same cells across animals? or is not fully penetrant or with variable expressivity across motor neurons? Depending on the phenotype, combining all motor neurons in a single group might not be the appropriate data representation. Additionally, it will significantly change the conclusions that can be derived from the data.

We thank the reviewer for this comment. In the revised manuscript, we have clarified that the *unc-3* mutant phenotype is fully penetrant (100%) with high expressivity in the same cells across animals (Results). This applies to both *unc-3* mutant populations of MNs (VD-like, VC-like).

For the VD-like population, ectopic expression of the VD terminal identity genes (*ser-2, oig-1*) is observed in all *unc-3* mutant animals (100% penetrance) (Figure 2A-B). To test whether this phenotype has high expressivity in the same cells across animals, we used fluorescent reporters (*ttr-39::mCherry, unc-47::mChOpti*), that are unaffected by loss of *unc-3* (Figure 2—figure supplement 1B), as landmarks for GABAergic nerve cord motor neurons (VD and DD) in wild-type (Figure 1—figure supplement 1), and *unc-3* (Figure 2—figure supplement 1B) mutants. Because the cell body position of cholinergic and GABAergic MNs is known in *C. elegans* and unaffected in *unc-3* mutants, we were able to unambiguously identify the *unc-3*-depleted motor neurons that acquire VD terminal identity features (VD-like). Through careful reporter analysis with single-cell resolution, we found the following MNs to consistently acquire expression of VD genes in *unc-3* mutants: DA4, DA5, DA7, DB5, DB6, DB7, VA3, VA4, VA5, VA6, VA9, VB4, VB9 (new Figure 2—figure supplement 1B).

For the VC-like population, ectopic expression of the VC terminal identity genes (*ida-1, glr-5, srb-16*) is observed in all *unc-3* mutant animals (100% penetrance) (Figure 2C-D). By performing the same type of single-cell analysis (new Figure 2—figure supplement 1B), we found that the following MNs consistently acquire expression of VC genes in *unc-3* mutants: VB7, AS2, AS3, AS4, AS5, AS6, AS7, AS8, AS9, AS10.

Lastly, we found that the VD-like and VC-like motor neurons represent distinct populations by generating *unc-3* mutants that carry a green marker for the VC terminal identity gene (*ida-1*) and a red marker for the VD terminal identity gene (*ser-2*). No significant overlap of the two markers was observed in *unc-3* mutants (Figure 2E-F).

At large, this work revolves around the combinatorial control of cell fate. It further dissects the unc-3/Hox dependent regulation of motor neuron fate. Activation of neuronal type-specific features and preventing the expression of different cell fates is a well-described transcription factor activity during chick and mouse motor neuron development. I am confused by the strong emphasis on "transcriptional switch". It is stressed as a novel insight but there is not tangible mechanistic evidence for it.

Reviewer #1 also commented on previous studies that describe activation of neuronal type-specific features and prevention of expression of different cell fates. Please see our sixth response to reviewer #1. In brief, the revised manuscript now discusses our findings in the context of these previous studies.

Moreover, our new experiments provided additional mechanistic evidence for the observed transcriptional switch in the targets of LIN-39. In brief, LIN-39 and UNC-3 bind on enhancers of terminal identity genes in cholinergic MNs of wild-type animals (ChIP-seq data for LIN-39 binding on these genes shown in Figure 5C, Figure 5—figure supplement 1). In the absence of UNC-3, we observed a dramatic decrease of LIN-39 binding on these genes by performing LIN-39 ChIP-seq in *unc-3* mutants (Figure 5C, Figure 5—figure supplement 1). Guided by the LIN-39 ChIP-seq data, we performed a detailed *cis*-regulatory analysis for *oig-1*, a gene normally activated by LIN-39 in VD neurons of wild-type animals and ectopically expressed in “VD-like” neurons of *unc-3* mutants (new Figure 8). We deleted *cis*-regulatory regions that are normally bound by LIN-39 in the context of *oig-1* transgenic reporter animals. This analysis strongly suggests that LIN-39 acts through distinct *cis*-regulatory elements (that contain consensus LIN-39 binding sites, Supplementary file 2) to activate *oig-1* expression in VD versus VD-like cells of *unc-3* mutants. Lastly, we re-wrote a section of Discussion entitled “*UNC-3 determines the function of the rate-limiting factor LIN-39/Hox in cholinergic motor neurons*” to describe better the mechanism in light of the new data we obtained during the revision process.

Moreover, the phenotypes are changes in gene expression with repression and activation of terminal genes, not a complete switch of fates as far as I can tell.

Yes, the phenotypes described in this manuscript concern changes in expression of terminal identity genes, whose protein products are critical for MN function (e.g., ion channels, neurotransmitter receptors, neuropeptides, etc.). In Results, we now clarify that the phenotype is not a complete fate switch (subsection “UNC-3 has a dual role in distinct populations of ventral nerve cord (VNC) motor neurons”). For example, the VD-like cells in *unc-3* do not acquire GABAergic neurotransmitter identity like the wild-type VD cells. Moreover, we mention in the Discussion subsection entitled“Limitations and lessons learned about the control of neuronal terminal identity”, that is not possible to evaluate whether the VD-like cells in *unc-3* mutants acquire morphological characteristics of wild-type VD cells due to the strong axon guidance defects observed in *unc-3* mutants. Lastly, we discuss the implications of this incomplete cell fate switch for the evolution of novel cell types (Discussion).

At the mechanistic level of gene regulation by transcription factors, there is no evidence of unc-3 or lin-39 switching binding targets or DNA motif preference.

Prompted by this comment, we conducted additional experiments that helped paint a clearer picture of the mechanism, now described in the revised manuscript. By using available LIN-39 ChIP-seq data, it is evident that LIN-39 directly binds to a host of terminal identity genes specific for cholinergic MNs (new Figure 5C, Figure 5—figure supplement 1), corroborating our genetic analysis. By performing ChIP-seq for LIN-39 in *unc-3* mutants, we detect that LIN-39 binding to these genes is dramatically decreased (new Figure 5C, Figure 5—figure supplement 1). Our new RT-PCR data, in *unc-3* mutants, on these cholinergic MN terminal identity genes further corroborated the ChIP-seq data (Figure 5—figure supplement 1). Together, these new data strongly suggest that LIN-39 and UNC-3 bind to *cis*-regulatory elements of cholinergic MN terminal identity genes and activate their transcription. In *unc-3* mutants, LIN-39 binding to these elements is decreased, presumably leading to increased availability of LIN-39 that results in activation of alternative identity genes (e.g., VD genes). Our new data support this possibility.

First, we quantified the levels of UNC-3 and LIN-39 proteins in cholinergic MNs, and found LIN-39 to be expressed at considerably lower levels compared to UNC-3, indicating that LIN-39 is a rate-limiting factor (new Figure 9A).

Second, we performed a comprehensive *cis*-regulatory analysis of the *oig-1* locus and found that LIN-39 acts through distinct *cis*-regulatory elements to activate *oig-1* in VD neurons versus VD-like neurons of *unc-3* mutants (new Figure 8).

In brief, LIN-39 and UNC-30/Pitx bind to the same *cis*-regulatory element of *oig-1* (called peak#3 in Figure 8A). This element is required for *oig-1* expression in VD neurons of wild-type and *unc-3* mutants. This element contains a number of weak LIN-39 consensus sites (based on p value, Supplementary file 2), suggesting that UNC-30 stabilizes LIN-39 binding on these sites. However, this element (peak#3) is not required for *oig-1* expression in VD-like cells of *unc-3* mutants (Figure 8A-B). Moreover, we identified a distal *cis*-regulatory element (300bp) of *oig-1* required for expression in VD-like cells. Interestingly, several strong consensus LIN-39 binding sites are found within this element (Supplementary file 2), suggesting that, in the absence of *unc-3*, LIN-39 no longer binds to cholinergic gene promoters and instead recognizes these strong consensus binding sites, presumably due to higher affinity to these sites.

We have revised the text (Results and Discussion) to include these new results, which are in complete agreement with our gene dosage experiments (Figure 9).

Additionally, transcription factor target switch by changing binding partners has been extensively described, even during motor neuron development.

Reviewer #1 made a similar comment. Please see our sixth response to reviewer #1.

In addition, we now present our findings in the context of previous studies describing transcription factor target switch by changing binding partners. Specifically, we mention in Discussion previous work in mouse and chick motor neurons, where Islet1, Lhx3, and NLI form a hexameric complex (called MN hexamer) necessary to activate MN identity genes. In V2 interneurons, Islet1 is not expressed, and Lhx3 and NLI form a tetrameric complex (V2 tetramer) that recognizes distinct DNA sequences compared to MN hexamer, thus activating V2-specific genes. However, as mentioned in our sixth response to reviewer #1 and Discussion, the underlying mechanism downstream of the MN hexamer and V2 tetramer involve a transcription repressor (Hb9, Chx10). Thus far, we found no evidence for a transcriptional repressor downstream of UNC-3.

Lastly, we cite two papers that involve transcription factor target switch by changing binding partners, one in *C. elegans* sensory neurons (Gordon and Hobert, 2015) and one in nascent mouse MNs in vitro (Rhee et al., 2016) (Introduction, second paragraph).

Finally, there is a trend to make claims that are not strictly supported by the data or a lack of precision describing phenotypes.

We thank the reviewer for this comment, as it helped us improve the clarity of the manuscript.

We now describe in more detail the *unc-3* and *lin-39* mutant phenotypes in the Results. Please, note the inclusion of new schematics summarizing the *lin-39* mutant phenotype in Figures 5D and 7F. Lastly, we do recognize that two incorrect claims were made in the original version. Our new data helped us understand better the molecular mechanism and, in the revised manuscript, we have corrected these claims as shown below:

1) LIN-39 is the rate-limiting factor (Results and Discussion).

2) Cholinergic motor neurons are not a uniform population, as they can be subdivided into multiple subtypes. The incorrect statement is removed from Introduction and Discussion.

In my opinion, this work has a few confirmatory results, the significant claim is not supported by the data and thus it does not provide a significant conceptual nor technical leap for eLife.

As mentioned above, the revised manuscript explains better why our study significantly advances our understanding of Hox gene function in the nervous system (Discussion). Moreover, an extensive set of new experiments (ChIP-seq for LIN-39 in *unc-3* mutants shown in Figure 5C, Figure 5—figure supplement 1; comprehensive *cis*-regulatory analysis of VD terminal identity gene *oig-1* shown in new Figure 8; quantification of LIN-39 levels shown in Figure 9A) helped clarify the molecular mechanism of action for UNC-3 and LIN-39. We hope that the revised manuscript now addresses the reviewer’s concerns.

Reviewer #3:This paper investigated the molecular mechanism used to determine and maintain terminal identity features of cholinergic motor neurons in the C. elegans ventral nerve cord. The authors showed dual function of the terminal selector UNC-3 (Collier/Ebf) in directly promoting identity features of cholinergic motor neurons and indirectly preventing expression of terminal features of three other ventral cord neuron types (VD, VC, CA). Through an unbiased genetic screen, the authors identified the Hox protein LIN-39 (Scr/Dfd/Hox4-5) to be required for ectopic expression of VD and VC features in unc-3 mutants. In summary, the data presented in the paper supports the dual role of UNC-3: UNC-3 and LIN-39 directly co-activate terminal identity features of cholinergic motor neurons; UNC-3 antagonizes the LIN-39's ability of activating terminal features of alternative neuronal identities.The paper is well written. However, some points should be addressed or clarified.- It has been previously shown that unc-3 and lin-39 in act synergistically to activate expression of cholinergic motor neuron terminal identity genes (Kratsios et al., 2017). This study identified a new role of unc-3 in antagonizing the LIN-39's function of activating terminal features of alternative neuronal identities. The authors should perform additional experiments to provide a mechanistic insight into how UNC-3 prevents LIN-39-mediated transcriptional switch for inducing expression of alternative identity features.

Our new set of experiments helped paint a clearer picture of the molecular mechanism underlying the dual role of UNC-3 and support the conclusion that LIN-39 is indeed the rate-limiting factor.

By using available LIN-39 ChIP-seq data, it is evident that LIN-39 directly binds to *cis*-regulatory elements of cholinergic MN identity genes (new Figure 5C), corroborating our genetic analysis. By performing ChIP-seq for LIN-39 in *unc-3* mutants, we found that LIN-39 binding is dramatically decreased to these gene loci (new Figure 5C). Our new RT-PCR data, in *unc-3* mutants, on these cholinergic MN terminal identity genes further corroborated the ChIP-seq data (new Figure 5—figure supplement 1). Together, these findings suggest that LIN-39 binding on cholinergic MN identity genes depends on UNC-3, raising the possibility that UNC-3 sequesters LIN-39 (or diverts LIN-39 activity) away from VD gene promoters. Supporting this scenario, ectopic expression of UNC-3 in VD neurons results in decreased expression of *lin-39*-dependent VD-specific markers (Figure 9C, Figure 9—figure supplement 2C).

In addition, we quantified the endogenous expression levels of *lin-39* and found it to be expressed at relatively low levels in cholinergic MNs (Figure 9A), further supporting the conclusion that LIN-39 is the rate-limiting factor. This conclusion is further supported by our gene dosage experiments for *unc-3* and *lin-39* (Figure 9B-C). For example, over-expression of LIN-39 in cholinergic MNs results in activation of VD-specific genes (Figure 9C). Together, our results suggest that the collaboration of UNC-3 with LIN-39 on cholinergic MN identity gene promoters exhausts the limited amount of LIN-39 in the cell and VD genes remain off. However, LIN-39 is released from cholinergic MN gene promoters in the absence of UNC-3 and activates VD genes.

Lastly, we now provide evidence that LIN-39 acts through distinct *cis*-regulatory elements to activate the same VD-specific gene (*oig-1*) in wild-type VD neurons versus VD-like neurons of *unc-3* mutants (new Figure 8, please also see our sixth response to reviewer #2). We have revised the text to include these new findings (Results) and also discuss them in the context of literature mentioned by reviewers #1 and #2 (Introduction and Discussion).

- The authors showed data to indicate that lin-39 and unc-30 act in parallel to co-activate VD terminal identity genes. However, the authors also showed that ectopic expression of VD markers is not dependent on UNC-30. The contradictory results should be discussed. Does the data suggest that lin-39 acts with yet-to-be identified factor(s) to activate ectopic expression of VD features in unc-3 mutants?

We thank the reviewer for this comment, which helped us clarify, in the revised manuscript, the role of UNC-3 and LIN-39 in VD neurons. First, we now include ChIP-seq data for UNC-30 and LIN-39 that show direct binding for both proteins on the *cis*-regulatory region of VD terminal identity genes (*oig-1, ser-2, flp-11*) (new Figure 8, Figure 8—figure supplement 1). Second, both UNC-30 and LIN-39 positively control expression of VD genes; however, there are some noteworthy target gene-specific effects.

*oig-1*: UNC-30 is absolutely required for *oig-1* expression. LIN-39 is also required, but milder effects were observed in *lin-39* versus *unc-30* mutants (Figure 7B). The strong *unc-30* effects precluded us from building the double mutant.

*ser-2 and flp-11*: Mutants for *lin-39* show stronger effects on *ser-2* and *flp-11* expression when compared to *unc-30* mutants. Double *lin-39; unc-30* mutants showed additive effects, suggesting that LIN-39 and UNC-30 act in parallel to activate *ser-2* and *flp-11* expression (Figure 7B, Figure 4—figure supplement 2, panel E).

We note that the strongest possible alleles (putative null) were used for our analysis [*lin-39(n1760), unc-30 (e191)*].

Lastly, over-expression of LIN-39 in cholinergic motor neurons (which do not express *unc-30*) is sufficient to induce expression of the VD marker *ser-2* (Figure 9B).

The aforementioned genetic analyses together with the LIN-39 over-expression experiment indicate that LIN-39 is both necessary and sufficient for inducing *ser-2*. Hence, in *unc-3*-depleted MNs (which do not express *unc-30,* but do express *lin-39*) ectopic expression of *ser-2* is observed in *unc-3 mutants (*Figure 2A-B). This ectopic expression is dependent on *lin-39* as evident by our analysis of *unc-3; lin-39 (-)* double mutants (Figure 4D). However, the *ser-2* ectopic expression was not abolished in *unc-3; unc-30* double mutants (Figure 4—figure supplement 1, panel A), since *unc-30* is not expressed in cholinergic MNs. We now describe better these findings in the Results and with a new schematic in Figure 7F.